# Cryostructuring of Polymeric Systems. 55. Retrospective View on the More than 40 Years of Studies Performed in the A.N.Nesmeyanov Institute of Organoelement Compounds with Respect of the Cryostructuring Processes in Polymeric Systems

**DOI:** 10.3390/gels6030029

**Published:** 2020-09-10

**Authors:** Vladimir I. Lozinsky

**Affiliations:** A.N. Nesmeyanov Institute of Organoelement Compounds, Russian Academy of Sciences, Vavilov Street, 28, 119991 Moscow, Russia; loz@ineos.ac.ru; Tel.: +7-499-135-6492

**Keywords:** cryostructuring, cryogels, cryostructurates, physico-chemical properties, macroporous morphology, applied potential

## Abstract

The processes of cryostructuring in polymeric systems, the techniques of the preparation of diverse cryogels and cryostructurates, the physico-chemical mechanisms of their formation, and the applied potential of these advanced polymer materials are all of high scientific and practical interest in many countries. This review article describes and discusses the results of more than 40 years of studies in this field performed by the researchers from the A.N.Nesmeyanov Institute of Organoelement Compounds, Russian Academy of Sciences—one of the key centers, where such investigations are carried out. The review includes brief historical information, the description of the main effects and trends characteristic of the cryostructuring processes, the data on the morphological specifics inherent in the polymeric cryogels and cryostructurates, and examples of their implementation for solving certain applied tasks.

## 1. Introduction

The preparative approaches based on the cryostructuring processes in the polymeric systems for the fabrication of new advanced materials that can be successively employed in various applied fields have acquired substantial interest for the last several decades. Thus, the amount of published scientific books, reviews, and papers within this topic has already exceeded 2400 [1], and the quantity of the respective intellectual property documents (international and national patents, inventor’s certificates, know-how, etc.) is so massive that it could hardly be estimated correctly. Another demonstration of the significance of this branch of polymer science is the impending publication of the Special Issue “Cryogelation and Cryogels”, which is in preparation now by MDPI [2]. Such interest by and large is due to the two main factors. First, a rather simple and commonly available freezing technique is used in such cryostructuring processes. Second, rather often the resultant polymer materials possess amazing properties.

In general, such materials could be classified as follows [3]:

I. Polymeric cryogels are generated when the gel-formation occurs exactly in the frozen system, and upon its thawing the formed 3D polymeric network becomes swollen in a defrosted solvent. In turn, the nodes of the three-dimensional polymer network of such cryogels can be stabilized by covalent, ionic, coordination and non-covalent bonds or a combination of them. A more detailed discussion of these systems is given in Section 2.

II. Cryostructurates are the macroporous polymer matrices, whose formation can be accomplished via two different schemes:

II(i). The first variant is the freezing of molecular or colloid solution of the non-gelling precursors followed by the removal of the frozen solvent by the techniques of freeze-drying or cryo-extraction. If it is required, additional tanning (cross-linking) of the sponge-like matrix thus prepared has to be done to make the material insoluble.

II(ii). The second variant is the freezing of the already prepared covalent or non-covalent (physical) gel with the consequent removal of the frozen solvent with the use of freeze-drying.

Sometimes, instead of the term “cryostructurate” the word “cryotexturate” is also exploited, mainly in the food technology [4].

In all these cases, final cryogels and cryostructurates possess the system of large pores, since the polycrystals of frozen solvent play a role of the porogens, and such macropores are, as a rule, interconnected [5,6,7,8,9,10,11]. The procedures for the preparation of these polymeric materials, the mechanisms of their formation, and the properties of the resultant cryogels and cryostructurates are the targets of numerous publications. Thus, the list of the wide-known reviews, including two fundamental books [12,13], contains more than 40 titles [5,6,7,8,9,10,11,12,13,14,15,16,17,18,19,20,21,22,23,24,25,26,27,28,29,30,31,32,33,34,35,36,37,38,39,40,41,42,43,44,45,46,47]. These are mentioned mainly to demonstrate the comprehensive scientific background of the R&D activities in this field.

Owing to the combination of similar specific porosity and wide possibility to adjust the physico-chemical characteristics of the various cryogenically structured matrices to the requirements of the goal application, polymeric cryogels and cryostructurates have been extensively examined and used in many practical areas. The following overview publications describe and discuss the generalized data related to the applied potential of similar polymeric materials in such fields as biomaterials (the highest quantity of the respective publications) [5,12,13,16,17,38,45,48,49,50,51,52,53,54,55,56,57,58,59,60,61,62,63,64,65,66,67,68,69,70,71,72,73,74,75,76,77,78,79,80,81,82,83,84], biotechnology [5,12,13,16,17,18,19,30,48,85,86,87,88,89,90,91,92,93,94,95,96,97,98,99,100,101,102], medicine [5,12,13,14,15,16,17,49,51,59,60,61,64,77,78,79,80,82,83,103,104,105,106,107,108,109,110,111], separation and purification techniques [5,12,13,19,31,81,89,92,93,97,112,113,114,115,116,117,118,119,120,121,122,123,124,125,126,127,128,129,130,131,132,133,134,135,136,137,138,139,140,141,142], chemical processes [5,29,142,143,144,145], food technologies [4,5,146,147], and construction and ecology engineering [5,16,148,149,150]. To the best of the author’s knowledge, in the scientific literature, there are no similar extended series of the reviews more or less touching various polymeric cryogels/cryostructurates and their applied potential. Of course, these areas do not at all embrace the exhaustive list of the examples that are reported in the experimental papers and patents dealing with the successful implementation of the various cryogenically-structured polymeric matrices. For instance, such interesting cryogel-based items as the materials for cleaning and restoration of old and modern artworks [151,152,153] or such exotic uses of the cryostructuring technique as the fabrication of fuel briquettes and gel fuels [154,155,156,157] can be mentioned.

In the last decades, the application of the cryostructuring processes for the preparation of materials based on natural or synthetic polymers featuring a desired set of properties, the study of such materials and their use are the subjects of scientific and applied activities in many countries [26]. The references to the above-indicated publications [2,3,4,5,6,7,8,9,10,11,12,13,14,15,16,17,18,19,20,21,22,23,24,25,26,27,28,29,30,31,32,33,34,35,36,37,38,39,40,41,42,43,44,45,46,47,48,49,50,51,52,53,54,55,56,57,58,59,60,61,62,63,64,65,66,67,68,69,70,71,72,73,74,75,76,77,78,79,80,81,82,83,84,85,86,87,88,89,90,91,92,93,94,95,96,97,98,99,100,101,102,103,104,105,106,107,108,109,110,111,112,113,114,115,116,117,118,119,120,121,122,123,124,125,126,127,128,129,130,131,132,133,134,135,136,137,138,139,140,141,142,143,144,145,146,147,148,149,150,151,152,153,154,155,156,157] demonstrate a widespread geography of similar works. Among the participants of the respective studies (very often interdisciplinary ones) are the studies from the A.N.Nesmeyanov Institute of organoelement compounds (IOEC), Russian Academy of Sciences [158], where the investigations of such processes and the materials thus prepared have been performed for more than 40 years. Many of the results that have been obtained during this period, especially those related to the molecular and submolecular mechanisms of the relevant processes, have a pioneering character and serve now as a scientific basis for the new applied elaborations. In this context, the goals of the given review are to describe the main directions of the previous and current fundamental, as well as applied studies in the IOEC; to discuss briefly the most principal experimental results; to emphasize certain general regularities on the influence of various factors on the cryostructuring efficiency; and to speculate a little bit about some practical prospects of such polymeric materials, as cryogels and cryostructurates.

## 2. Short Historical Remarks

The empirical data on the formation of various cryogenically-structured polymeric matrices have been known since long ago [5,26]. These observations were mainly connected to the frozen storage of food systems [4] and with the wide-pore fungous appearance of the polymeric texturates that were being formed as a result of the freeze-drying processing, when such technology had been developed [6,23]. In fact, the real basic studies of the cryostructuring processes of the polymer systems commenced in the 1970s (e.g., see [5,26,48,49,159,160,161,162]). The first studies performed in the IOEC within this topic were realized at approximately the same time, when the target systems under the investigation were the food-type biopolymers [146,163,164]. These explorations were carried out in the framework of programs dealing with the chemistry and physical/colloid chemistry of food substances [165]. Subsequent development of the studies of the cryostructuring processes in this institute was in progress with respect of two general directions: (i) food-related systems; and (ii) studies of the mechanisms of the cryogenically-structured materials formation with simultaneous testing of the diverse variants for the implementation of elaborated cryogels and cryostructurates in practice. The cumulative data on such basic and applied works are collected separately in Table 1 and Table 2, respectively, where the information is located in a retrospective order—from the early publications to the recent ones.

The jointing core for all these previous and current activities is the application of cryostructuring approach for the preparation of one or other polymeric matrices that, if such tasks exist, could be used for solving of certain applied problems. In general, for such cases, a common technical operation is the freezing process. During this stage, the liquid–solid phase transition occurs, and the biggest fraction of the initial solvent is crystallized. As a result, the formed polycrystals (ice in case of aqueous systems) fulfill the function of porogens for the final cryogenically-structured polymer matrices, and the initial precursors are concentrated in the space of so-called unfrozen liquid microphase (UFLMP) [397]. Such cryoconcentrating effect is known to be one of the main driving forces for the cryotropic gel-formation phenomena [5,9,35,66,142]. That is why the used solvent has to be crystallizable rather than vitrifying one.

As concerns the cryotropic gel-formation, in the course of studies performed in the IOEC, it was shown (for the majority of the below-listed systems, it was done for the first time) that, analogously to the traditional gels formed in the liquid media at the temperatures above the freezing points, the cryogels (I-type in the Introduction) can be prepared originating from the following types of the precursor systems [2,5,26,167]:

I(i). Solutions of the low-molecular monomers or high-molecular monomers (macromers) plus initiators: the target chemically cross-linked cryogels are formed via the polymerization processes (*2, 5, 8, 15, 25, 29, 32, 54, 58, 59,* and *66* in Table 1).

I(ii). Solutions of the low-molecular monomers suitable for the formation of covalently cross-linked polymeric networks via the polycondensation reactions (*2* in Table 1).

I(iii). Solutions that contain the system “polymer(s) + cross-linking chemical reagent(s)” (*1–4, 7, 16, 52, 68, 70,* and *78* in Table 1); the additional variant is the crosslinking function fulfilled by the irradiation [18,28,32,48].

I(iv). Polyelectrolytes plus crosslinking counterions or chelators; the junction nodes in the 3D-networks of the resultant cryogels are linked by means of the low-dissociating ionic or coordination bonds (*26, 27* in Table 1).

I(v). Solutions of the polymeric precursors capable of undergoing the cool-induced sol-gel transition with the formation of physical junction nodes within the bulk of the resulting 3D network (*6, 9–11, 17–19, 22, 23, 28, 30, 31, 34–36, 38–42, 44, 46, 47, 51, 57, 61–63,* and *72–74* in Table 1).

I(vi). Colloidal dispersions that are capable of gelling (*37, 55,* and *65* in Table 1).

Taking into an account these types of the precursor systems, the cryogels, analogously to the conventional gels, with respect of the nature of interchain links in the network nodes can be classified as covalent (chemical), ionically cross-linked, and non-covalent (physical) gel matrices [5,9,26]. Certainly, the systems of a “mixed” kind also exist, when the combinations of covalent, ionic, and non-covalent interchain links reside within the common cryogenically-structured polymeric material. For instance, this is often so for the cryogels based on such heterofunctional macromolecular precursors, as proteins [163,164,175,207,215,257,259,262,272,306,317,381,386] or amphiphilic synthetic copolymers [247,249].

In their turn, the precursors of the II(i)-type cryostructurates (see the Introduction) could be virtually any synthetic or natural polymer soluble in the respective crystallizable solvents. The same is also true for the cryostructurates of the II(ii)-type; the only requirement is that the pre-formed polymeric network must be swollen in order to freeze it prior to the removal of the frozen solvent.

In addition, if the above-listed initial feeds contain the additives of the insoluble dispersed filler(s), cryogenic processing will result in the formation of the composite cryogels and cryostructurates. The feed compositions and the cryogenic processing conditions used for the preparation of such filled materials are also presented in numerous examples in Table 1 and Table 2. Certain features of similar cryogenically-structured particular composite are discussed below. Beyond these variants, in the case of the sponge-like cryogenically-structured matrices that possess the system of a capillary size (cross-section of tens and hundreds microns) interconnected wide pores, there is the possibility to soak the polymeric carrier with some suspensions of the micro- or nanoparticles that, owing to capillary forces, will penetrate into the matrix bulk. Subsequent drying of thus filler-loaded cryogels or cryostructurates will yield the composite materials (e.g., see [68,272,388,389,391,395]).

Therewith, the investigations of the mechanisms of the processes participating in the formation of various cryogels and cryostructurates, as well as the revelation of the key factors affecting the properties and macroporous morphology of these polymeric materials, are of primary importance. It is clear that the information obtained during such studies (Table 1) served as a reliable basis for the scientifically-grounded implementation of such materials in various areas of practice. In turn, the majority of similar “applied” R&D (Table 2) has been accomplished in a productive cooperation of the researchers from IOEC with the professionals from the respective branches of science and technology. It primarily relates to the biomedical and biotechnological fields (see the reviews [5,66,67,68,88,89,90,92,94,100,101,112], and the corresponding examples are collected in Table 2).

## 3. The Main Effects and Trends Characteristic of the Cryostructuring Processes

### 3.1. Specifics of the Macroporous Morphology of Polymeric Cryogels and Cryostructurates

As emphasized above, the macroporosity together with a system of (as a rule) interconnected pores are the characteristic features of the cryogenically-structured polymeric matrices [5,6,7,8,9,10,11,12,13]. Such interconnections of macropores are formed as a result of the contacts between the growing solvent polycrystals during the freezing of the system, providing that no special techniques for the directed freezing are implemented. Upon the thawing stage, these contacts are transformed into “interlinks” between the macropores. The micrographs in Figure 1 exemplify some morphological peculiarities of the macroporous (pore size up to ~10 μm), as well as supermacroporous (pores from several tens to several hundreds of micrometers) cryogels and cryostructurates. The respective materials, the 3D network of which is crosslinked via non-covalent (Figure 1A–D,I), covalent (Figure 1E–G), or ionic (Figure 1H) bonds, have been prepared from the molecular solutions of monomeric (Figure 1E,F) or polymeric (Figure 1A–D,G–I) precursors or from the colloidal dispersions (Figure 1D) being frozen either in aqueous (Figure 1A,B,D,E,G,H) or in organic (Figure 1C,F,I) media. Since these micrographs are from different papers, the scales of the images are not the same. Nonetheless, as a whole, the images, along with the heterophase macroporous structure of such freeze-fabricated matrices, testify to the versatility of the cryostructuring approach with respect of its high preparative potential for the obtaining of various cryogels and cryostructurates with a very diverse “architecture” of the pore space and the polymeric phase properly, i.e., the pore walls.

The size (cross-section) and shape of the large pores that are, in fact, the replicas of the porogen matter—the polycrystals of frozen solvent—depend on numerous factors, the major ones found to be as follows [5,9,10,16,66,94]:

First: Chemical nature, solubility, and concentration of the precursors affect the fraction (volumetric percentage) of the initial solvent, which can be frozen-out at a particular cryogenic processing temperature. In turn, the amount of the frozen solvent polycrystals will define the total (macro)porosity of the final cryogels or cryostructurates. Thus, Figure 1A shows the macroporous structure of the physical PVA cryogel prepared by the freezing of the concentrated (100 g/L) aqueous solution of the polymer, which is able to hold a large amount of the unfreezable solvate water. As a result, an average size of the macropores in this sample was 2.7 μm, and its total (macro)porosity was 53.1% [232]. In turn, Figure 1B presents the cobweb-like morphology of the agarose-based cryogel, which is physical gel matter, as well. The sample was prepared by the freezing of the less-concentrated (30 g/L) solution of the polymer, so the larger amount of the solvent could be frozen-out. As a result, the cross-section of the gross pores was 50–250 μm and the total (macro)porosity exceeded 97%; since, during the cryotropic gel-formation, the polymer was concentrated in the “narrow” ULMP layers between the ice particles, after the system thawed, it yielded rather thin walls of macropores of the final spongy cryogel [237].

Second: Chemical nature and cryoscopic properties of the used solvent are the factors strongly influencing the quality of the feed system to be supercooled upon chilling and on the morphology of the porogens—frozen solvent polycrystals. These factors will define the size and shape of the macropores of the final cryogels or cryostructurates. For instance, one can see very pronounced differences in the macroporous morphology of the polymeric cryostructurates, whose formation occurred in the media of freeze-crystallized naphthalene (Figure 1C), frozen water (Figure 1G), and freeze-crystallized DMSO (Figure 1I). No doubt, such differences were stipulated mainly by the type of the used solvents, more exactly by the geometry of their crystals.

Third: Often such additives as the foreign solutes and the disperse fillers are able to exert significant influence on the crystallization of the liquid precursor system, thus affecting the porosity characteristics of the resultant cryogenically-structured materials. For example, proteinaceous cryogel (Figure 1F) formed by the freezing of the aqueous feed solution, which, along with a high-molecular precursor (serum albumin), also contained dissolved guanidine hydrochloride and had the gross pores of the oblong ellipsoid shape with the minor axis average size of 15–50 µm and the major axis up to 100–400 µm and even longer. At the same time, in the case of similar matrices prepared in the presence of urea additives, such pores were of roundish appearance [272], similar to the pores in the Ca-alginate cryostructurate exemplified in Figure 1H. In general, the additives of such solutes as the low-molecular electrolytes, because of their either salting-in or, in contrast, salting-out abilities, were shown to affect markedly the macroporous characteristics of the respective cryogels and cryostructurates [203,219,246,261,269,387]. The same note is also related to the influence of various chaotropes and kosmotropes introduced in the feed solutions prior to the system freezing [194,195,207,257,267,268,272]. In addition, a pronounced influence has been detected when the surfactants were introduced in the composition of feed system [227,230]. The latter effects were exhibited because of the surfactant-caused decrease in the surface tension on the solid–liquid interphase, thus influencing the size and shape of the forming solvent polycrystals. Beyond that time, the effects connected to the particulate additives are often manifested in a rather heterogeneous macroporosity of such cryogenically-structured samples (e.g., see [215,273,274,317]). However, in some cases, a very interesting regular texture can be observed. For instance, it is so for the cryostructurates based on the freezing-sensitive latexes, when the resultant material (Figure 1D) appeared to possess an astonishing “geometrically-ordered” textile-like microstructure [254].

Fourth: If simultaneous to the liquid–solid phase separation (solvent crystallization) some additional phase transformations occur, such processes also affect the macroporous morphology of the cryogenically-structured final polymeric matrices. For instance, when the feed solution, along with the gel-forming macromolecular precursor, additionally contains other non-gelling polymer, the cryoconcentration of both these polymers can cause the liquid–liquid phase segregation inside the volume of UFLMP because of the thermodynamic incompatibility of different polymers in the common solvent. Such combination of the in-parallel-proceeding processes has resulted in the formation of different kinds of pores in the final cryogels. Thus, the image in Figure 2A shows the appearance of the PVA-based sieve-like disk prepared by the freeze–thaw processing of the mixed aqueous solution of two macromolecular compounds: the gelling component (PVA) and the gum arabic, which is unable to gel under the freeze–thaw conditions [251]. The gross through-hole pores with the cross-section of around 0.1–1.0 mm can be clearly seen even with the naked eye. Optical microscopy reveals (Figure 2B) that such gross pores are interconnected; they have been generated by the non-gelling gum arabic-rich microphase, which had been washed-out then by the experimenters after the sample defrosting. In turn, the gel matter proper, i.e., the continuous phase of the heterogeneous material, contains roundish pores of 1–10 µm in cross-section (Figure 2C). Hence, definite hierarchy of the porosity is observed. Namely, at least two distinct types of macropores differing significantly in their size are seen to exist in such spongy cryogel prepared by combining the liquid–liquid phase separation and the cryotropic gelation processes.

There are also some other examples of the cryogenically-structured gel systems that acquired “additional” gross pores as a result of the liquid–liquid phase segregation occurring in parallel with the formation of the freezing-induced pores. Examples are the cryogels prepared on the basis of the PVA-solutions in DMSO (Figure 2D) with the additives of low-molecular alcohols, in particular methanol (Figure 2E,F) [264]. In this case, as the DMSO is crystallized first upon cooling, the unfrozen fraction of the system is segregated into two phases: the concentrated DMSO solution the polymer and the low-concentrated DMSO solution in methanol. The formation of the PVA cryogel occurs in the former phase, and the inclusions of the latter phase perform as “additional” porogens. The higher is the initial methanol concentration, the larger are such gross pores (compare Figure 2E,F).

Fifth: A somewhat different sort of the additional phase transformation takes place when the initial system to be cryogenically structured includes foreign solutes possessing the desolvation abilities with respect of the gel-forming polymer. The example is a partial coagulation of the gelling polymer, as in case of the physical cryogels prepared by freeze–thaw processing of PVA aqueous solutions (rather than the previous example with the PVA/DMSO-systems), which also contained the additives of simple aliphatic alcohols, e.g., methanol (Figure 2G–I) [252]. Since upon moderate freezing of such solutions the ice is crystallized, MeOH concentration in the UFLMP grows and reaches a threshold level at which some fraction of PVA begins to coagulate with the formation of the “secondary” fine mesh consisting of the PVA microfibers (dark strands in Figure 2H,I). The size and morphology of such “secondary” mesh are dependent on the MeOH initial concentration and freezing temperature. For the sake of comparison, in Figure 2G, the microstructure of the methanol-free PVA cryogel prepared under the identical conditions is also shown.

Sixth: Very significant factors that influence substantially the specifics of the macroporous morphology of the polymeric cryogels and cryostructurates are the thermal conditions of all stages of the cryogenic processing. These factors are numerous [5,6,7,8,9,10,11,35,38,45,63,66,94,125,144], including:

-the material kind and thermal conductivity of the vessel (mold), in which the system to be frozen is located;

-the quality of the inner surface (smooth or rough) of the respective vessel;

-the ability of the initial liquid system to be supercooled prior to the solvent crystallization;

-the cooling rate during freezing;

-the existence (or not) of the directed heat sink;

-the freezing temperature itself;

-the frozen storage duration;

-the rate of the frozen samples heating upon their defrosting;

-the number of freeze–thaw cycles.

While, the first seven of these points are of significance for virtually all types of polymeric cryogels and cryostructurates, the latter two parameters are of special importance for the physical cryogels such as the ones based on PVA, starch polysaccharides, locust bean gum, and some other polymeric precursors [5,9,14,15,16,17,89,101,175,179,180,189,208,210,211,217,218,219,220,225,232,233,234,243]. As demonstrated, e.g., for the PVA cryogels, their formation mainly proceeds over the pre-melting temperature range upon the frozen system thawing [9,16,180,190,193,214,232]. Therefore, the slower is the heating rate, the longer does the gelling system reside at the temperatures of the most intensive gel-formation, when the polymer concentration in the UFLMP is still high enough, and, at the same time, some fraction of the solvent is already liquid, thus providing the conditions for the segmental/molecular mobility of the chains and their efficient interactions [16,189,216,218,232,233,234]. In this context, multiple freeze–thaw cycles create a possibility for the gelling system to pass several times through the above-indicated temperatures of the most intensive gel-formation, thus facilitating the increase in the yield of the process. With that, every subsequent solvent crystallization step (commencing from the second run) occurs within the space of the already formed “primary” pores rather than in the volume of free liquid as it happens upon the first freezing. The physical stresses arising in the course of such repeated solvent crystallization cause widening of the macropores in the respective physical cryogels and result in the growth of the density and rigidity of the polymeric walls of the macropores [9,14,15,16,17,22,233].

As concerns the temperature regimes of the cryogenic processing, the lower is the temperature inside the freezer and the higher is the cooling rate, the smaller, as a rule, are the solvent polycrystals and, consequently, the smaller would be the macropores in the resultant cryogenically-structured polymeric matrices [5,9,10,22,23,24,25,66,125,170,177,178,179,180,184,232,233,234,235,236,237,257,262,263,270,272]. Moreover, with the lowering of the temperature, the higher amount of the solvent is frozen-out, so the total (macro)porosity, i.e., the overall volume of such pores in the resulting cryogel or cryostructurate, is increased [5,9,10,186,190,193,197].

In this context, the temperature profile, which is used in the course of the freezing of the initial liquid system, should also be noted. For instance, it concerns the cases, when so-called “low-temperature quenching” technique is employed in order to freeze the sample quickly, and only then to rise the temperature of the thus frozen system up to the desired level. Such freezing procedure can result in the formation of specific macroporous morphology in the final cryogels or cryostructurates. The spatial architecture of the gross pores and their polymeric walls in such matrices as if combines the structural “motives” characteristic of the matrices prepared at both the low and the sub-zero temperatures [5,9,177,184,197,222,237].

Such factor of the cryostructuring processes, as the duration of the samples incubation in a frozen state, and its influence on the properties and structure of the resultant polymeric matrices is in close association with the dynamics (velocity) of the structuring phenomena within the volume of the UFLMP [5,9,16,177,179,180,185,186,218]. If these processes occur slowly, a prolonged frozen storage is preferable to reach the higher yields, while, in the case of the fast cryostructuring, the extended incubation is not required, especially taking into an account the possibilities of the re-crystallization of the non-deeply-frozen solvent. In turn, similar re-crystallization is able to affect the amount and size of the macropores in the final cryogels and cryostructurates [232,234]. In this connection, the necessity of the quantitative characterization of such structural parameters of these materials as their general porosity, an average cross-section of the pores, the pore size distribution, etc. are of significance. Both manual measurements and computer-assisted approaches have been used for the mathematical treatment of the images obtained with the aid of various types of microscopy (optical [170,171,172,206,225,230,231,232,233,234,237,238,240,241,243,244,245,246,250,251,252,256,257,258,259,261,262,263,264,265,267,268,269,270,272,380,383,384,390,393], laser confocal [369,370,371,372,392], and SEM [173,174,177,179,181,182,184,196,202,205,207,229,235,236,274,339,340,370,372,388]). Therewith, in the case of cryogels, the most informative are usually the data acquired for the intact swollen samples rather than for the dried ones, since any drying of the swollen gel matter causes its compacting and, as a consequence, distortion of the shape and the size of both the pores and their polymeric walls.

Summing up the data obtained in the IOEC on the influence of the above-discussed parameters of the cryogenic structuring processes on the specifics of the macroporous morphology of various covalent, ionic, and non-covalent cryogels, as well as cryostructurates, it is necessary to conclude that virtually all of these factors are more or less able to exert the effects. Hence, in each particular case, certain studies are required to reveal the dominant factors and establish the major trends of their influence as dependent on the other conditions of the respective cryogenic processing.

### 3.2. Some Peculiarities of the Mechanisms of the Cryostructuring Processes in Different Polymeric Systems and the Influence of These Peculiarities on the Physico-Chemical Properties of the Resulting Cryogels and Cryostructurates

In fact, all studies carried out at the IOEC as well as other studies on the preparation of the diverse cryogenically-structured polymer materials have always included the evaluation of various physico-chemical characteristics of the respective samples, and, very often, the dependence of their properties on the preparative conditions. The detailed discussion of the results of such studies and of the general effects thus observed for different variants of the cryostructuring processes have been presented in several comprehensive reviews published by the author of the present paper and by other scientists [5,6,7,8,9,10,11,12,13,14,15,16,17,20,22,27,28,29,40,44,63,66,82,94,125,142,143]. Therefore, only the key selected features of the cryostructuring mechanisms are considered below.

(1) One of the primary effects inherent in the cryotropic gel-formation processes was found to be a perceptibly lower initial concentration of the precursors at which it was possible to prepare cryogels in the non-deeply-frozen systems in comparison with the gelling of the same substances in the non-frozen liquid media [5,9,169,170,171,172,173,175,176,177,184,185,186,220,226,237]. The main reason for such an effect is the cryoconcentration of the respective precursors in the volume of UFLMP, when their specific concentration is increased by several times, at least, as dependent on the freezing temperature. However, such a decrease in the critical concentration of gelation of the precursors upon the formation of cryogels is an apparent effect, since the cryotropic gel-formation occurs in a significantly more concentrated system than the feed one [5,9,16,187,190,193,197]. The manifestation of such effects has been registered for different types of the precursor systems used for the preparation of the polymeric cryogels. It is so for the formation of the covalent cryogels by means of the polymerization of the low-molecular monomers (precursor systems I(i) and I(ii) in Section 2) [172,173,177,184,197] or via the chemical cross-linking of the macromolecular compounds (precursor systems I(iii) in Section 2) [169,170,171,176,185,186]. It is also so for the non-covalent cryotropic gelation of the molecular (precursor systems I(v) in Section 2) [175,187,190,220,226,237,383] or colloidal (precursor systems I(vi) in Section 2) [215,273,274] solutions. Similar apparent decrease in the critical concentration of the precursors upon the cryotropic gel-formation was observed not only in aqueous, but also in the organic frozen media (e.g., see [173,175,176,383]), i.e., the effect is common for such processes.

(2) Another effect closely connected to the cryoconcentration phenomena is the acceleration of the cryotropic gel-formation processes taking place over a certain range of negative temperatures [5,9,16]. Thus, for the case of the formation of the polymerization-type cryogels, the fact thar reacting the gel-point in the moderately-frozen reaction system is faster than the gel-formation of the same but unfrozen liquid system for the first time has been shown by the examples of poly(acrylamide) cryogels [172,173,177,184]. For instance, during the redox-initiated copolymerization of acrylamide with *N*,*N*’-methylene-bis-acrylamide (molar ratio of 30:1) in their 3% aqueous solution (*25* in Table 1) at +20 °C, the gel-point (the gel-fraction yield of about 10%) was reached after ~90 min, whereas, in the frozen system at −20 °C, it took ~15 min, i.e., an about six-fold acceleration was observed [184]. Similar character effects were found to be inherent in the gel-forming systems of the “polymer + crosslinker” type. This case can be exemplified by the gelation of thiol-derivative of poly(acrylamide), when the cross-linking of macromolecules occurred via the SH-groups oxidation by the dissolved air oxygen with the formation of the interchain disulfide bridges (*16* in Table 1). It turned out that the rate of thiol oxidation in frozen at −15 °C aqueous polymer solution exceeded the rate of the same process in the liquid system at +15 °C by a factor of around 5 [186]. As for the non-covalent cryotropic gelation, such acceleration effects were registered, as well. The most known examples are the PVA-based cryogels: concentrated aqueous solutions of this polymer undergo the sol-to-gel transition at low positive temperatures only after prolonged (several days) incubation with the formation of very weak and low-melting (30–35 °C) gels, while a single freeze–thaw cycle allows obtaining rather elastic cryogels, the fusion temperatures of which are markedly higher (60–80 °C) [16]. Another example is the freeze–thaw-caused gel-formation of the locust bean gum aqueous solutions (*42* in Table 1). As at the positive temperatures such solutions transform into very weak gels only after 2–3 months, the freeze–thaw processing yields physical cryogels in several hours and without the use of any cross-linking chemicals [220]. Of course, the main reason for all similar acceleration effects is the cryoconcentration of the gel-precursors in a small volume of the UFLMP of the respective partially (incompletely) frozen molecular or colloidal solutions [5,9,10,16,19,22,27,35,45,63,66,90,112,142].

(3) At the same time, below a definite boundary of negative temperatures (this is a specific point for every particular system), the rate and extent of the gel-formation efficiency commence to be suppressed. As a result, the dependences of these parameters on the temperature of the cryotropic gelation processes are usually of a bell-like character, thus indicating the competition between the promoting and hindering factors, which have been found to be as follows [5,9,10,30]:

The above-discussed cryoconcentrating effects are first related to the promoting factors that powerfully facilitate the cryotropic gel-formation.

In the case of the exothermic reactions involved in the formation of the spatial polymeric network of the respective cryogels, the lowering of the process temperature assists the heat withdrawal and has to shift the equilibrium toward the final “products”; thus, this mechanism is also related to the favoring factors.

When the covalent cryogels are formed by the heterolytic reactions in the medium of a non-deeply-frozen organic solvent (e.g., the case of the preparation of the polystyrene-based cryogels *7b* in Table 1), the lowering of the temperature causes the increase in the dielectric constant of the UFLMP in comparison to the polarity of the initial liquid medium. Such an effect facilitates the charge separation, thus accelerating the similar reactions.

The additives of certain solutes capable of exhibiting the kosmotropic (antichaotropic) properties are the substances that can efficiently promote the non-covalent cryotropic gel-formation, in which the appearing intermolecular junction nodes are stabilized by the H-bonding of such polymeric precursors, as starch polysaccharides, PVA, locust bean gum, etc.. For example, the rigidity and heat endurance of the aqueous PVA cryogels formed in the presence of the simple salting-out inorganic electrolytes (NaF, KCl, Na_2_SO_4_, Na_3_PO_4_) (*31, 57,* and *75* in Table 1) or such kosmotropic organic substances, as oligoethylene glycols, trehalose, or hydroxyproline (*23, 28,* and *73* in Table 1) were increased significantly in comparison to the additive-free samples. In turn, for the PVA cryogels prepared in the frozen DMSO medium, the unexpected conversion of the effects caused by the additives similar to urea or guanidine hydrochloride has been observed. These chaotropes (in aqueous media) exerted the kosmotropic-like influence on the physico-chemical properties of the resultant cryogels (*74* in Table 1). Such phenomenon, which promotes the formation of physical cryogels, was shown to be stipulated by the worsening of the thermodynamic quality of the DMSO with respect of the PVA because of the H-bonding of the solvent and the molecules of the above-indicated additives [267,268].

The deceleration of the thermal mobility of the gel-precursors within the space of UFLMP with the lowering of the process temperature is related to one of the substantial mechanisms negatively affecting the gel-formation performance.

Other significant “actors” that interfere with the cryotropic gelation are the very high viscosity of the UFLMP and the formation of the nodes of the 3D polymeric network inside of this unfrozen phase (e.g., see [187,190,193]), both hindering the interactions that participate in the cryostructuring processes.

Contrary to the promoting influence of the kosmotropes on the formation of the physical cryogels, the solutes possessing the chaotropic properties inhibit this type of gelation. In particular, such effects were observed when the additives of the salting-in low-molecular inorganic electrolytes (e.g., LiCl, NaSCN, NH_4_Cl) or organic chaotropes such as urea or guanidine hydrochloride have been introduced in the PVA aqueous solutions prior to their cryogenic processing. The resultant cryogels were weaker and less thermoresistant in comparison to the additive-free samples of the same polymer concentration [203,246,261].

As a result of the competition of the above-listed factors the dependences of the rate and efficacy indicators for the gel-formation in various moderately-frozen polymeric systems on the temperature of the cryogenic processing have the bell-like character; when in the vicinity of the maxima, the intensity and gelation yield can very often exceed considerably the same values reached at positive temperatures [5,9,10,90,170,171,176,177,180,184,186,187,218,225,232,234,235,262].

Additional interesting exhibitions of similar bell-like temperature dependences peculiar to the processes in frozen systems are the effects related to the linear cryopolymerization of the respective unsaturated monomers [398,399,400,401,402,403,404,405]. In these cases, such extreme dependences were found regarding the reaction rate, the yield of the target polymeric products, and, most importantly, the molecular weight of the resultant polymer. At that, the chain length of the polymers formed in the moderately-frozen systems was, as a rule, markedly higher than for the polymers prepared originating from the same initial reaction solutions but at the temperatures above their freezing points [398,399]. Such effects have been registered for the cryopolymerization processes in both aqueous [398,399,400,401,402,403] and organic [403] frozen media, thus demonstrating a common character of the effects. For instance, upon the redox-initiated polymerization of the 0.5-M acrylamide aqueous solution at +20 °C the resultant poly(acrylamide) had a molecular weight of about 412 kDa, while the polymer synthesized in the frozen at −12.5 °C (the top point onto the respective bell-shaped temperature dependence) had a molecular weight of about 7400 kDa, i.e., it was higher by a factor of about 18 [399]. For the thermoresponsive poly(N-isopropylacrylamides) prepared in liquid (+20 °C) and frozen (−10 °C) aqueous media (initial monomer concentration: 0.5 mol/L), the molecular weights were 5060 and 8590 kDa, respectively [401]. The example of an analogous effect in an organic medium is the case of the *N*,*N*-dimethylacrylamide polymerization in liquid and crystallized formamide [405]. Thus, the molecular weight of the polymer synthesized in the 1-M formamide solution of the monomer at +22.9 °C was about 220 kDa, and for the poly(*N*,*N*-dimethylacrylamide) formed at −19.6 °C in the frozen reaction system, i.e., by >40 °C lower, the molecular weight was found to be 810 kDa. Similar molecular-weight-associated effects probably also influence the formation and properties of the polymerization-type cryogels, but the studies on such an influence so far have not been performed at IOEC or by other explorers.

### 3.3. Filled (Composite) Cryogels and Cryostructurates

Various composite cryogels and cryostructurates containing different dispersed fillers entrapped inside the polymeric matrix have been the matters of both basic studies and applied elaborations (for reviews, see [5,12,13,22,39,64,65,68,74,87,88,89,90,91,92,94,95,96,97,98,99,100,148,149,150]). The interest in such macroporous and supermacroporous composite materials is mainly stipulated by their high potential for the use in practice. The presence of the fillers, their characteristics and concentration affect the property–structure relationships in the respective cryogenically-structured composites. In particular, it concerns the chemical nature and microstructure of the fillers, their compatibility with the continuous phase and the type of the particulate matter (the solid particles, flexible small fibers, soft deformable particles like the foreign particulate gels or microbial cells, droplets of immiscible liquids, gas bubbles, etc.).

In general, the following major pathways for the preparation of such cryogenically-structured filled materials are feasible and have been applied:

(i) The preparation of the suspension of the disperse filler in the feed solutions followed by the freeze–thaw structuring of the system, thus yielding the target composite [5,13,22,39,65,68,85,86,87,88,89,90,91,92,94,95,96,97,98,99,100,101,148,149,150]. Here, the specific case is the use of the mycelial fungi spores as the fillers, when the composite matrix, after its formation, is placed in the growth nutritional medium for the mycelium development. As a result, the germinating hyphas pierce the space of the macropores in the polymeric carrier and reinforce the final biocatalytically-active composite material [5,86,89,90,92,94,95,96,100,278,282].

(ii) The preparation of the feed solution, which contains the mixture of the precursor of the polymeric matrix to be formed, the precursor of “future” filler, and the agent capable of causing the formation of a filler. Then, this mixture is quickly frozen to provide the conditions for the cryotropic structuring with a simultaneous formation of the discrete phase [231,252]. One of the examples of such systems is the preparation of the silica-filled PVA cryogels by combining the freeze–thaw gel-formation of the aqueous PVA solutions and hydrolytic polycondensation of the tetramethoxysilane (TMOS) added to the feed solution just prior to the freezing of the reaction mixture (*50* in Table 1). In this case, the inductor of the filler formation is water, which hydrolyzes the TMOS with a further formation of the silica microparticles entrapped in the matrix of the resultant composite cryogel [231].

(iii) The preparation of the feed solution, which contains the precursors of the polymeric matrix to be formed and the soluble precursor of the “future” filler, followed by freeze–thaw processing of such mixed solution. Subsequent treatment of the resultant polymeric matrix with an agent, which is able to diffuse into the porous bulk, causes the formation of the filler discrete phase inside the continuous phase of the already prepared cryogel or cryostructurate [258,259,260,261]. This variant can be exemplified by the procedure used for obtaining of the filled PVA cryogels in which the discrete phase is the microparticulate chitosan (*69* in Table 1). In such a case, the primary complex cryogel is prepared by cryogenic processing of the solution, which contains the mixture of the gel-forming polymer, i.e., PVA, and the non-gelling polymer—chitosan hydrochloride (or acetate). At the next stage, the resultant gel matter is incubated in the ammonia vapors to cause the transformation of the dissolved chitosan salt into the insoluble chitosan base. Final rinsing of this material with water removes all solutes from the porous matrix, while the microparticles (2–5 μm) of the precipitated chitosan remain inside the bulk of the composite [258].

(iv) A separate variant is the preparation of the hybrid “inanimate-alive” composites that are of interest for the tissue engineering field. In this case the wide-porous spongy cryogels or cryostructurates are initially fabricated. Further, the sponge-like scaffold is seeded with primary cells, the subsequent propagation of which inside the pore space of the polymeric carrier gives rise to the desired hybrid material filled with the in situ grown biomass [5,13,45,59,62,63,64,67,70,71,72,73,77,79,82,94,98].

All these procedures have been implemented for the preparation of the composites under consideration, the peculiarities of their properties have been explored, and the operational characteristics of numerous functional materials thus obtained have been examined in the processes of interest. The majority of the studies accomplished in IOEC dealt with the filled PVA cryogels (e.g., see *12, 20, 24, 33, 43, 45, 48, 49, 50, 56, 60, 64, 67, 69,* and *75* in Table 1 and *5, 6, 8, 10, 11, 13, 14, 17, 20–22, 25, 26, 30, 32, 33, 37–39, 41–43, 46, 50, 56, 57,* and *68* in Table 2), although the cryostructured composites based on other polymers have also been involved in the respective investigations (*13, 14, 21,* and *54* in Table 1 and *7, 55,* and *60* in Table 2). The particular fillers used in these studies were as follows:

-solid inorganic and organic particles (*33, 45, 48b, 50, 60, 64, 67,* and *77* in Table 1 and *17* and *60* in Table 2);

-semi-solid deformable gel particles (*24, 48a*, and *69* in Table 1 and *32* in Table 2);

-microbial cells (*12–14, 20, 21,* and *33* in Table 1 and *5–8, 10, 11, 13, 14, 20–22, 26, 26, 30, 33, 37–39, 42, 43, 46, 50,* and *55–57* in Table 2);

-micro- and nanofibers (*33b* in Table 1 and *68* in Table 2);

-oil-in-water emulsions (*54* and *56* in Table 1);

-gaseous micro-bubbles (*43* and *49* in Table 1).

With that, upon the respective studies, a series of rather specific physico-chemical effects coupled with the interactions of the polymeric phase and various fillers has been observed and their mechanisms have been revealed. Some of these effects turned out to be common for covalent as well as non-covalent freeze-structured composites, whereas some effects were characteristic of only individual polymer–filler pairs.

According to a mutually-known concept, the disperse fillers, with regard to their influence on the physico-chemical properties of the resulting polymeric composites (including the gel-type ones), are classified as “active” (reinforcing the materials), “inactive” (without pronounced influence), and “worsening” fillers (causing the weakening of the materials) [406,407]. Therefore, the effects induced by the various fillers that were entrapped in certain cryostructured polymer matrices are discussed below in view of this classification.

Of course, such usual filler-associated influence, as the growth of the mechanical strength of the composites with an increase in the content of the solid filler, providing good compatibility of discrete and continuous phases, has been observed also for the composite cryogels, in the most cases for the PVA-based ones. For instance, it was so for similar cryogels formed in a frozen aqueous medium with simultaneous its filling with the small mineral particles of sufficiently hydrophilic silica [205,225], titania [205], marble, and coquina [206]. At the same time, when the filler was more hydrophobic and only limitedly compatible with the continuous phase, the particulate matter formed agglomerates and did not cause marked reinforcement of the resultant composites [206]. However, if the composite PVA cryogels filled with hydrophobic particles (not only inorganic, but also the organic ones like the cross-linked polystyrene [224]) were prepared in the medium of the frozen organic solvent DMSO, the phase compatibility was good enough. As a result, the particulate matter was uniformly distributed within the cryogel bulk, and the reinforcement effect was pronounced [206]. Thus, these examples highlight the importance for the properties of the final composites of the filler and the cryogel compatibility, which facilitates the tight adhesion between the discrete and continuous phases.

In addition, the extent of the compatibility or the incompatibility of the phases can also be controlled by the ionic interactions even in the cases when the continuous gel phase is uncharged. Thus, such phenomena have been observed for the composite PVA cryogels filled with the particulate ion-exchange resins (*48* in Table 1). For instance, when the latter ones were the salt forms of the ion-exchange Sephadexes (particles of the ionic derivatives of the cross-linked dextran hydrogel), the less rigid composite cryogels were formed as the concentration of the entrapped ionite was increased. However, when the strong anionites in the (OH^–^)-form or strong cationites in the (H^+^)-form were used as the fillers, a substantial increase in the rigidity and the heat endurance of the resultant composites took place [228]. These effects were associated with the additional ionic bonding between the continuous and dispersed phases. Such Coulombic interactions were the consequence of the partial ionization, respectively, deprotonation or protonation, of the PVA-belonging hydroxyl groups. Since the pK_a_ values of the OH-groups of secondary alcohols in an aqueous medium lie in the 13–14 range, the (OH^−^)-forms of such tetraalkylammonium-carrying strong anion-exchange resins, as the OAE-Sephadex or the QAE-Amberlite at their sufficiently high concentrations are able more or less to deprotonate the PVA. The cryoconcentrating effects in the UFLMP of the moderately-frozen filler-PVA suspensions facilitate this mechanism. As a consequence, the formation of the ionic bonds such as …-N(C_2_H_5_)_3_^+^**···**O^−^-…between the filler and the gel matrix can occur. These de novo ionic pairs are fixed by the cryotropic gelation due to a drastic decrease in the mobility of the polymer chains in the formed gel network, thus resulting in an additional increase in the mechanical strength and the heat endurance of the prepared composites [228,229].

The principal significance for the properties of the cryogenically-structured composite materials was shown to be the porosity of the filler phase [196,205,206,261]. When the size of the pores in the particles of the dispersed filler is wide enough for the penetration of the precursor molecules, the subsequent cryogenic structuring will proceed not only within the space of the continuous phase, but also inside the discrete phase, thus yielding the formation of the network-in-network structures. In such a case every filler particle turns out to be as if “sewn-in” the overall 3D polymeric network and plays a role of the macroscopic junction node. In particular, such morphological feature has been revealed by the SEM studies of the composite PVA cryogels filled with the particles of various porosities [196,205]. In turn, the presence of similar nodes tightly-bound with the continuous phase of the resultant composites gives rise to their pronounced strengthening.

Other interesting types of cryogenically-structured composites are those filled with the dispersions of the immiscible liquid fillers (*54* and *56* in Table 1 and *41* in Table 2). Thus, when the synthesis of the thermoresponsive poly(N-isopropylacrylamide)-based cryogels have been performed in the frozen oil-in-water emulsions of the surfactant-stabilized tetradecane or various natural oils, the resultant sponge-like composites contained microdroplets of the hydrophobic liquid being entrapped in the volume of the macropores. Such materials were able to squeeze the oil phase through the system of the interconnected gross open pores upon the volumetric shrinking of the polymeric phase in response to the temperature exceeding the LCST point. In contrast, the same, with respect to the chemical composition, oil-filled microporous hydrogels synthesized at a room temperature were unable to perform similar squeezing of the oil phase, though the heating-induced collapse of the matrix occurred, and this process “conserved” the hydrophobic liquid inside the bulk of the shrunk composite rather than caused the oil plug flow [241,242].

The next examples of cryogenically-structured composites that contain the “immobilized” oil-type dispersed fillers are the PVA cryogels formed by the freeze–thaw treatment of the respective oil-in-water (more precisely, aqueous PVA solution) emulsions [234,244,348]. The mechanically-forced leakage of the hydrophobic liquid to the outer aqueous medium was found to be specific for such composites: a gradual release of the oily matter could be induced by the cycling compression of these composite cryogels. At the same time, even a prolonged incubation simply under the water layer did not yield any marked oil release [244]. Since the PVA-cryogel-based matrices are considered to be very promising materials for the biomedical applications [5,12,13,17,22,37,51,52,53,54,55,56,60,65,75,77,82,110], e.g., for the elaboration of the artificial cartilages [107,108,109,111], the possibility for such polymeric systems to be filled with the medical-oil-soluble drugs that can be released slowly under the alternating mechanical impacts, appears rather prospective.

Some interesting peculiarities of the properties and the functionality have been found to be inherent in the gas-filled polymeric cryogels and cryostructurates [36,148,221,230]. Such materials can be prepared either by the mechanical foaming (e.g., by whipping) of the feed solutions followed by their cryogenic processing thus yielding the air-filled composites, mainly the PVA-cryogel-based ones (*43* and *49* in Table 1), or by the chemical formation of the small bubbles as a result of the gas-emitting reactions with the participation of the respective substances incorporated in the system to be freeze–thaw structured [407,408]. The properties of similar composites depend, along with the cryogenic processing conditions, on the characteristics of the initial fluid foams, in particular, on the foam stability (the foam half-life) and the so-called “foaming capacity”—the ratio between the volume of the foamed system and initial volume of the polymer solution before the foaming procedure [221]. Since the intrinsic density of the gas-filled PVA cryogels is less than the density of water, such composites are capable of floating in water for a long time (one month, at least) without any changes in the buoyancy. This result evidently means that the air entrapped inside the gas bubbles of the foamed cryogels was not replaced by water, e.g., via the simple diffusion mechanism.

Since the spatial heterophase “architecture” of the diverse cryogels and cryostructurates significantly affects their integral properties, certain features of the influence of the various fillers on the macroporous morphology of some cryogenically-structured polymeric matrices are briefly discussed at the end of this section. For the sake of a better comparison, such features are represented here by the examples of the composite PVA cryogels, the microstructure of which is shown in the micrographs of Figure 3, and as the reference sample the image in Figure 1A answering to the unfilled PVA cryogel is used. Although the scales of all these images which were taken from different papers (e.g., Figure 1 and Figure 2) are not the same, the main specifics of the polymeric matrices of interest can be well distinguished. The examples in Figure 3 are related to the composites that contain the following sorts of dispersed fillers: the solid mineral particles (Figure 3A–C), soft particles of coagulated “foreign” polymer (Figure 3D), the microdroplets of liquid oil (Figure 3E,F), and air microbubbles (Figure 3H,I).

Thus, upon suspending the well-compatible silica particles in the aqueous PVA solution and subsequent cryogenic processing of this system, the filler becomes virtually uniformly distributed within the resultant macroporous gel matrix (Figure 3A). At the same time, when instead of the unmodified silica its hydrophobized C_18_-derivative (the adsorbent for the reverse-phase chromatography) was used as a filler, its particles aggregated, and the large fraction of the final polymeric material remained unfilled (Figure 3B). Therefore, a general character of the structural morphology of these two composite PVA-cryogels was essentially different [225]. All the more markedly distinctive was the microstructure of the filled PVA cryogels also containing SiO_2_ particles (Figure 3C), but the latter ones were formed in situ as a result of the hydrolytic polycondensation of TMOS introduced in the feed polymer solution to be gelled cryotropically (*50* and *77* in Table 1). The specific peculiarity of such composite, along with the entrapped solid filler particles, was the presence of the thick strands (stained with Congo red dark lamellar structural elements in the black and white image) within the polymeric phase. It was shown that the latter were the PVA microfibers formed by the local coagulation of the polymer caused by the methanol liberated during the TMOS hydrolysis in the frozen gelling system [231,252].

Another example of the composite PVA cryogel also containing the in situ generated discrete phase is the system mentioned in Section 3.3. (ii), in which the small (2–5 μm) filler particulate matter (Figure 3D) was formed through the transformation of the dissolved chitosan hydrochloride to the water-insoluble chitosan-base (*69* in Table 1). In this case, such a large amount of the soft polymer coagulate particles is seen to be uniformly distributed inside the material, as the porous structure of the gel phase is almost un-distinguishable at the resolution of the given image. Thanks to the large number of the filler particles, their tiny size, and, as a consequence, gross value of the available surface, such chitosan-containing composite PVA cryogel was able to absorb the heavy metal ions, e.g., Cu^2+^, in the quantities close to the equimolar ones relatively to the chitosan amino-groups [258].

The micrographs in Figure 3E,F are related to the case of the composite PVA cryogel filled with the Vaseline oil dispersed droplets, i.e., the filler is the water-immiscible hydrophobic liquid (*56* in Table 1). The former image is the light microscopic appearance of the initial emulsion, i.e., the oil dispersed in the aqueous PVA solution, and the latter image is the thin section of the resultant composite cryogel. The diminution in size of oil droplets entrapped in the gel matrix as compared with the initial emulsions was observed. Such an effect is most likely explained by the disintegrating action of crushing and shear stresses arising upon the system freezing and the growth of ice crystals [244].

The bottom row of images in Figure 3 shows the microstructure of three gas-filled PVA cryogels (*43* and *49* in Table 1) prepared by the cryogenic processing of the fluid foams that were of equal foaming capacity, but of somewhat different composition. The left sample did not contain any foreign stabilizer of the initial whipped foam; in the middle sample, the fluid foam was stabilized by the non-ionic surfactant; and, in the right sample, the cationic surfactant was added upon the foaming. The main morphological feature of such gas-filled cryogels is the presence of the pores of two types: very large pores (air bubbles with a diameter of tens and even over a hundred micrometers) and smaller pores (with the cross section of several micrometers) inside the gel phase of these heterophase systems [230]. The surfactant-free sample (Figure 3G) and the cryogel prepared in the presence of non-ionic surfactant (Figure 3H) were not so different, whereas the gas-filled composite prepared in the presence of the ionic surfactant (Figure 3I) differs significantly from each other in the number, shape and size of large pores, as well as in the morphology of the interface around the air bubbles. These features are stipulated by several reasons. One of them is a partial disintegration of the initial foam in the course of the cryogenic processing. Seemingly, the bubble–liquid film system, due to a considerable decrease in the surface tension at the bubble–PVA solution interface in the presence of surfactants, is deformed under the stresses arising upon the ice crystallization. Such distorted foam structure is fixed by its freezing and the PVA gelation. An interesting aspect of the influence is the variation in the microstructure of the gel phase both in the bulk of these macroporous systems and directly near the surfaces of the air bubbles entrapped in the cryogel matrix. In the samples without and with the added non-ionic surfactant, the gel phase is a highly porous system having the extended pores with the cross sections of 1–2 μm; the gel near the surface of air bubbles is denser than far from bubbles (Figure 3G,H). In turn, the addition of ionic surfactants to the initial composition changed the structure of the gel phase, where the pores were enlarged up to 4–5 μm in cross section (Figure 3I). With that, the bubbles inside such composite had the “pitted” surface, which cannot exist in any fluid foams. However, the formation of the cryogel in the bubble walls prevented a complete disintegration of the bubble shell system, thus fixing specific intermediate structure of the boundary layers “pierced” by the sharp ice microcrystals. These effects testify to a stronger influence of the ionic surfactants (both cationic and anionic [230]) comparing with the non-ionic ones on the properties and porous morphology of the foamed PVA cryogels. Most probably, this tendency, along with other factors, is also connected to the efficiency of the polymer–surfactant association processes capable of affecting the resistance of the bubble-polymeric phase interfaces against the freezing-induced physical stresses.

## 4. Applied Potential of Some Cryogenically-Structured Polymeric Materials Developed in the IOEC

As frequently happens in the fields of polymer and material sciences, the motivations in the IOEC for the early studies related to the cryostructuring processes were also connected to certain applied tasks. In Section 2 it is indicated that such tasks concerned the freeze-caused structuring of the food-type biopolymers. Furthermore, subsequent studies, especially those directed to revealing the fine chemical and physical mechanisms of the processes resulting in the formation of various cryogenically-structured polymeric matrices, allowed the development of a large family of cryogels and cryostructurates based on both synthetic and natural polymers. Many of them turned out to be suitable for the practical applications, and the diagram in Figure 4 schematically describes the respective areas. Some essential examples of such materials and the methods of their preparation are briefly considered in the present section.

### 4.1. Food-Related Systems

Frozen storage of food products is commonly known to be one of the fundamental approaches to food preservation, which has been used by mankind for many centuries. At the same time, the investigations of the processes that occur in similar frozen systems and, moreover, the exploration of the cryogenic structuring phenomena, as well as their application for the preparation of the new food forms (e.g., sublimated meal [409]), are considerably “younger” [4]. The R&D efforts performed in IOEC within the latter topic were related to three main directions (Figure 4): (i) the controlled cryostructuring of the food components or their mixtures in order to fabricate the freeze-textured edible products; (ii) the studies of the individual peculiarities inherent in cryostructuring of the food-related biopolymers, and (iii) the research on the specifics of the sorption–desorption interactions of the odorant compounds that have been incorporated into the biopolymeric matrices during the cryogenic processing.

In the course of these studies, it was found that freezing–incubation frozen–defrosting of the paste-like dispersions of the proteinaceous coagulates allowed to prepare the macroporous cryogels that could be then adjusted to the organoleptic properties and taste of real food products [146,163,164,273,274]. For instance, it was found that the cryostructuring of the paste-like coagulates of myofibrillar krill proteins resulted in the formation of the thermally-irreversible macroporous cryogels, the texture of which closely resembled the fish and crustacean tissues (*2* in Table 2). The subsequent culinary treatments of such cryogels gave rise to the obtainment of rather delicate foodstuffs [410].

It is evident that only knowing the main physico-chemical regularities of a process of interest and the factors influencing the final results, it is possible to develop a reliable and well-reproducible technology. Therefore, the data obtained upon the detailed studies of the cryostructuring mechanisms in the simplified (comparing to real foodstuffs) model systems were able to serve as a scientifically-grounded basis for the applied elaborations. These models were the systems that contained individual food-related biopolymers or their mixtures. In this exact context, it is worth to consider the investigations of the cryogenic structuring in such systems as the moderately-frozen aqueous solutions of starch polysaccharides (*39, 41, 46,* and *55* in Table 1), locust bean gum (*42* in Table 1), agar and gelatine (*6* in Table 1), ovalbumin (*34* in Table 1), etc. In these studies, many significant facts and trends have been established. For instance, the universal character of the non-covalent cryostructuring processes has been shown for the polymeric precursors capable of gel-forming via the intermolecular hydrogen bonding [175,220]. In addition, the key role of intermolecular hydrophobic interactions has been demonstrated for the cryostructuring of the globular proteins such as ovalbumin [207]. In the case of the mixed amylose–amylopectin systems (*41* in Table 1), the promoting influence of the freeze–thaw treatment on the polymer–polymer association, as well as the manifestation of some synergism in the mutual interaction of these polysaccharides, have been revealed [219].

In addition, the interesting examples of the food-related cryogels and cryostructurates are the starch-based ones that are able functioning as the carriers of the food-grade odorants, e.g., the aroma-compounds being the constituents of certain natural oils (*35* in Table 1). Such cryogenically-structured edible polysaccharide matrices can efficiently hold various aroma-compounds and gradually release them in the aqueous media. Thus, similar systems are of interest for the preparation of such foodstuffs as instant coffee, dried soups, etc. With that, the basic research data really facilitate the elucidation of the most optimum compositions of such products and conditions for their fabrication.

### 4.2. Cryogenically-Structured Materials of Biotechnological Purposes

Biotechnology, along with the food-related topics, was one of the first areas, where the polymeric cryogels and cryostructurates attracted attention in a view of their possible application [5,13,18,26,48,85,86,87,88,89,90,91,92,93,94,95,96,97,98,99,100,101]. First, such interest concerned the high-performance carriers for the immobilization of molecules and cells (mainly, microbial ones). It was because the cryostructuring technique allowed the preparation of the macroporous matrices that are able to secure the hindrance-free supply of the substrates and the rejection of products/metabolites, respectively, to and from the matrix-entrapped biologically-active “actors”. Table 2 contains the examples of similar immobilization carriers elaborated at the IOEC and in a collaboration with the biotechnologists who examined the resultant solid-phase biocatalysts in various processes.

Certainly, the type of porosity (macroporous or supermacroporous) and properties of each particular carrier depend on the task to be solved. Basically, such polymeric matrices should possess chemical and biological stability, good enough physico-mechanical characteristics that provide a long exploitation life of the respective materials. The macroporous morphology of the carrier is highly desirable in order to warrant a free mass transfer of the soluble substances and even nano/micro-particulate matter. The properties of many polymeric cryogels and cryostructurates answer well to these requirements [5,12,13,16,90,94,98,112].

Among the macroporous cryogenically-structured matrices the PVA-based cryogels were most often used by the researchers from the IOEC as the carriers of the immobilized molecules and whole cells. Since the chemical structure of PVA includes a large amount of the hydroxyl groups, there is a lot of possibilities for the introduction of a wide variety of reactive anchor functions that could be utilized further for covalent grafting of both low- and high-molecular-weight compounds to such carriers [5,13,16,66,87,92,94,411]. In particular, this approach was successfully implemented for the preparation of various immobilized biocatalysts based on different enzymes covalently-attached to the PVA-cryogel matrices (*31, 34,* and *47* in Table 2). The approach was also applied for the immobilization of virus-specific antibodies upon the elaboration of the respective immunosorbents (*29* in Table 2), as well as it was used for the attachment of the low-molecular ligands when preparing the bioaffinity sorbents for cells separation (*51* in Table 2). For instance, it was shown that, because of the multipoint binding of the enzymes to the carrier by means of stable chemical bonds, the biocatalytic macromolecules acquired an increased resistance to the denaturating action of polar organic solvents [94,321,322,323,324,325,326,327,328,329,330,331,332]. This quality allowed similar immobilized biocatalysts to efficiently operate in such media without a drastic loss of activity, as it frequently takes place for the same enzymes in their solutions or dispersions in the environment of organic solvents.

Another interesting possibility to immobilize the enzymes in the PVA-cryogel matrices is the entrapment technique (*32* and *68* in Table 2). This is a two-step procedure: initially, the insoluble form of the enzyme is prepared (e.g., it could be the so-called cross-linked enzyme aggregates [412] or the enzymes tightly adsorbed onto some insoluble support particles [413]), and then the resultant dispersion is suspended in the PVA solution followed by its cryotropic gelation. Such technique makes it possible to load final immobilized biocatalyst with a markedly higher amount of the enzymes in comparison to the covalent attachment [333,334]. However, the true choice of a more suitable variant of the immobilization scheme depends, of course, on the level of the residual biocatalytic activity of a particular enzyme.

In turn, the entrapment procedure is a rather convenient technique which has been developed for the immobilization of whole microbial cells in the carriers based on the PVA cryogels [16,85,86,87,88,89,90,91,92,94,95,96,97,98,99,100,278]. Owing to the combination of the outstanding operational characteristics and macroporous morphology, such gel carriers are among the best ones used for this purpose. The PVA cryogels possess high mechanical strength (up to 50–100 kPa) and heat endurance (up to 75–85 °C), they are virtually not subjected to an abrasive erosion even in the reactors with intense stirring, the macroporous structure of the carrier facilitates free diffusion of any dissolved substrates and metabolites. The preparative procedure is not sophisticated: the cell biomass is dispersed in the PVA solution, after which the cryogenic processing leads to formation of the composite—the cells entrapped in the matrix of the PVA cryogel (*5* in Table 2). The resultant immobilized biocatalyst can be fabricated in any desirable forms: blocks, sheets, tubes, beads, etc.; and for the preparation of the bead-shaped particles special cryogranulating setups have been elaborated (*18* in Table 2). Using this approach, a variety of different microbial cells (psychrophilic, mesophilic, and thermophilic) has been immobilized in the PVA-cryogel-based carriers: bacteria (*5, 6, 11, 13, 14, 20–22, 25, 26, 39, 42, 50*, and 57 in Table 2), yeasts (*20* in Table 1 and *5, 10,* and *30* in Table 2), filamentous fungi (*12* in Table 1 and *5, 8, 33, 37, 43, 46,* and *56* in Table 2). The respective immobilized biocatalysts were able to perform such functions, as simple mono-enzyme biotrasformations of definite substrates (*5, 6, 13, 14, 26,* and *57* in Table 2), the biosynthesis of either low-molecular (*11, 21, 22, 25, 30, 33, 37,* and *38* in Table 2) or macromolecular products (*12* in Table 1 and *10, 46,* and *56* in Table 2), the biosensing of certain analytes (*20* in Table 2), and the decomposition of undesirable pollutants (*39, 42, 43,* and *50* in Table 2). Thus, this wide set of possible processes testifies to a very high applied potential of the biocatalytic systems consisting of the microbial cells entrapped in the PVA-cryogel-based carriers.

Supermacroporous (wide-porous) cryogenically-structured spongy matrices have also been actively used as the immobilization carriers for the molecules and cells, especially for the processes performed in the columnar flow-through reactors [12,13,21,30,31,90,93,94,95,112]. In these cases, the flow of the input liquid is able to pass through the system of the capillary-size interconnected pores in such carriers, thus it is not necessary to prepare the particulate form of the intercolumn bed, which can be cryogenically fabricated simply as a wide-pore cylinder exactly inside such a reactor [5,177,237,339,340,366]. If required, the immobilization step can also be carried out under the flow-through conditions, and it was shown to be possible both for the molecules (enzymes, bioaffinity ligands, etc.) (*12, 36, 48, 49,* and *51* in Table 2) and for the whole cells (*13* in Table 1 and *48* in Table 2), since the cross-section of the gross pores in such matrices is high enough. The entrapment technique was also applied to the immobilization of the microbial cells in the spongy carriers of such type, if the cells were introduced to the precursor system to be structured cryogenically (*13* and *21* in Table 1 and in *7*
Table 2). As a result, the cells turned out to be caught within the thin walls of the macropores of the resultant composite sponge (e.g., see TEM images in [182]).

However, a more extensive development had a biotechnological direction related to the application of various wide-pore cryogels and cryostructurates in the bioseparation processes, i.e., upon the specific detection, isolation, and purification of the diverse biological molecules and bioparticles (protein bodies, plasmids, viruses, cell organelles, and whole cells) [5,12,13,90,93,94,95,98,112,113,114,115,116,117,118,119,120,121,122,123,124,125,126,127,128,129,130,131,132,133,134,135,136,137,138,139,140,141]. Among the numerous experimental studies, there are two frequently-quoted pioneering publications [339,340] that served as a real basis for further development of new materials and processes in the bioseparation field. The latter works dealt with the synthesis and implementation of the ion-exchange and bioaffinity derivatives of the poly(acrylamide)-based wide-pore cryogel for the separation of microbial cells and enzymes, respectively (*35* and *36* in Table 2). In the further studies, the same principle was repeatedly applied with success by various researchers using different cryogenically-structured polymeric matrices.

In addition, the sponge-like cryogels and cryostructurates have proven themselves to date very well as the high-efficient scaffolds for the 3D culturing of human and animal cells, since the spatial “architecture” of the large pores (e.g., see Figure 1B,H,I) and their walls turned out to be well-suitable for seeding, adhesion, and then proliferation of various cell lines, including stem cells followed by their subsequent transformation to the desired type of tissue cells [13,19,45,53,54,55,57,58,59,60,61,62,63,64,70,71,72,77,78,79,80,82,83,106]. In this field, the chemical nature, physico-chemical properties and porosity of the scaffolds are the key factors strongly influencing (of course, in a combination with the properties of the cells) the final biological results. Therefore, the scaffolds fabricated on the basis of the non-toxic biopolymers are, as a rule, more preferable than the synthetic matrices because of a potential danger of the generation of the undesirable substances upon the slow bio-induced decomposition of the latter materials. The next significant point is the presence of the recognition sites for cells onto the inner surface of the pore walls, what is necessary in a view of correct adhesion and spreading of the calls for their normal growth and functioning. If the primary framework of a certain scaffold does not contain such sites, those can be additionally coupled to the matrix, as it was done, e.g., by the covalent grafting of the gelatin macromolecules to the pore walls of the spongy agarose cryogels (*53* in Table 1 and *44* in Table 2) or Ca-alginate cryostructurates (*53* and *58* in Table 2). Since primary amino acid sequence of the gelatin chains is very close to the sequence of natural collagen, such modification of the scaffolds facilitated the affinity of the cells to the carrier’s material and assisted their growth [237,238,239,240,354,355,369,370,371,372,380]. On the other hand, the preparation of the collagen-based (*15* in Table 2) and gelatin-based (*59* in Table 2) cryogenically-structured spongy scaffolds is also possible, providing guaranteed sterility of the final materials, inasmuch as the raw matters are very often contaminated by microbes. The latter problem has been shown to be solved in the case of the gelatin scaffolds by performing the cryostructuring process in the DMSO/gelatin solutions followed by subsequent treatments performed in the ethanol medium (*65* in Table 2). Such procedure resulted in the sterile wide-pore scaffolds (Figure 1I) suitable for the 3D culture and transformation of stem cells [382,383,384].

An interesting variant of the application of the cryostructured scaffolds (*45* in Table 2) was realized upon the experiments with the so-called permissive cells, i.e., the cells capable of serving as the hosts for the propagation of certain viruses. It was found that such three-dimensional culturing of the virus-infected cells allowed increasing the amount of the resulting virus material by a factor of 3–5 as compared to the traditional monolayer cell culture, ones more showing promising potential of the cryogenically fabricated polymeric matrices for the purposes of cell technologies.

### 4.3. Polymeric Cryogels and Cryostructurates of Biomedical Interests

The wide-porous biocompatible 3D scaffolds fabricated via the cryostructuring technique are also of importance with respect of their potential application as bioartificial materials in the medical practice [13,59,62,63,64,67,71,72,77,78,79,80,82,83,84,106]. Some cryogels and cryostructurates (e.g., *53* in Table 1 and *58* in Table 2) have been elaborated for these aims in the IOEC in collaboration with biomedical professionals. Such matrices showed good results during the in vivo testing, in which the respective “cells-scaffold” constructs were implanted in the organisms of the laboratory animals. For instance, when the cryopreserved fetal liver cells seeded into the spongy alginate-gelatin scaffold (which was additionally coated with an additional immunoprotective alginate-gel-based shell) were implanted to the omentum of rats with hepatic failure, the operation resulted in a significant improvement of the hepatospecific parameters of the blood serum and positive changes of the liver morphology [380]. Thus, it was demonstrated that similar bioartificial constructs are the promising systems for the development of bioengineered liver equivalents.

Owing to their macroporosity, the sponge-like cryogels and cryostructurates, in particular those fabricated from the polymers permitted for the medical application, can be used as covers on wounds and burns, as well as in the drug delivery systems (Figure 4). In the former case the attractive quality is the high absorption capacity of such wide-pore matrices with respect of blood and exudates, in the latter case–the high loading possibilities of similar materials with regard to medicines or antimicrobial agents [52,59,66,68,83,84]. For example, the Ca-alginate-based cryostructurates (*26* in Table 1) formed by the freeze-drying of the Na-alginate aqueous solutions with a subsequent ionic tanning of the resultant sponge with saturated ethanolic solution of CaCl_2_ were able to retain stability in the physiological media for at least three weeks. In addition, the material was able to absorb a large volume of liquid and to release antimicrobial substances loaded into the biopolymeric matrix upon its preparation. The loading procedure itself is rather simple: the matrix is immersed into the drug solution, swells, and then is dried using an appropriate technique. Analogously, other sponge-like biopolymeric cryogels and cryostructurates, including those loaded with drugs or bactericides, are also of interest as covers on wounds/burns and the drug delivery carriers. Some of the examples of such cryogenically-structured materials elaborated in the IOEC are the absorbing/releasing sponges based on alginates and pectins (*26* in Table 1 and *3, 63,* and *69* in Table 2), chitosan (*27* and *76* in Table 1 and *69* in Table 2), globular and fibrillar proteins (*68, 70, 71,* and *78* in Table 1 and *15, 61, 64, 65,* and *69* in Table 2). Thus, the cryogels prepared from the serum albumin (*61* in Table 2) possess the very useful quality of exhaustive biodegradability inside the mammalian organisms. Therefore, these drug delivery cryogels loaded with antibiotics have been proposed as the efficient antibacterial sponges for the chemotherapy of the infected wounds [386]. At the same time, if it is required to decelerate the biological decomposition of such matrix, its chemical modification (e.g., by succinylation) can be used (*78* in Table 1) to decrease the affinity of the respective proteolytic enzymes in relation to similar proteinaceous substrate.

In one recent study, the sponge-like cryogenically-structured matrices fabricated from chitosan, alginate, or serum albumin were used as the carriers for the peptide bioregulators (*67* in Table 2). Such loaded sponges have been implanted in the model bone defect in rats. At the end of these experiments, 14 days after the surgery passed, it was revealed that the bioartificial implantates were effective as osteoconductors due to their impact on the osteoblast precursors. All types of the bioregulator-loaded matrices promoted active bone repair, which manifested in the restoration of the dense bone tissue, formation of the bone marrow, and the recovery of osteons. In contrast, as for the animals that received the implantation of reference sponges without the bioregulator, only the formation of the imperfectly dense fibrous tissue and spongy bone was observed [393]. The studies in this direction are in progress now.

In turn, macroporous cryogels such as the PVA-based ones are also known to possess a wide applied potential as the materials of biomedical interests; especially it concerns the orthopedics [49,60,65,107,109,110,111,396], the cardio-vascular surgery [105], the systems of controlled drug release [14,15,16,17,56,75,108], the so-called phantoms for the verification of NMR tomographs [103], and the medical ultrasound apparatus [104]. In this context, the data on the influence of the preparation conditions and the presence of different additives on the characteristics of the final PVA cryogels are of a key significance, since only knowing the mechanisms of such influence one can reach the desirable properties of the target gel material. The studies performed at the IOEC with respect of these cryogels have evidently demonstrated the sophisticated multifactor character of the structure/property relationships as dependent on the molecular parameters of the PVA itself and its initial concentration (*9, 10, 36,* and *51a* in Table 1), on the nature and composition of the used solvent (*6d, 19,* and *72–74* in Table 1), on the freeze–thaw conditions (*10, 11, 17–19, 22, 38, 40, 51,* and *63* in Table 1), on the properties and amount of the various additives (*20, 23, 24, 28, 30, 31, 33, 43, 47–50, 56, 60–62, 64, 67, 69, 72–75,* and *77* in Table 1), etc. As a result, and also taking into an account the data of many other authors (e.g., [14,15,17,27,37,49,51,56,60,65,69,75,76,107,111]), there are no serious problems nowadays to consciously vary the characteristics of these cryogels and adjust their properties to the desirable ones. It could concern the physico-mechanical characteristics of the concrete PVA cryogel, as well as its absorbing/desorbing properties, i.e., the ability to play a role in the controlled drug release gel system. As mentioned above, the pore size in the PVA cryogels is large enough for a free penetration of any solutes [411], but, if the gel-loaded substance interacts with the polymeric matrix by means of some adsorption mechanisms, a delay in the release can be observed. For instance, in the cases of PVA cryogels formed in the presence of amino acids of general formula H_2_N-(CH_2_)*_n_*-COOH (where *n* = 1–5) (*62* in Table 2), such phenomenon took place with an increase in the length of the oligomethylene chain. Its hydrophobic interactions with the carbochain core of the PVA were assumed to be the most probable reason for the delay effect.

Some other promising directions of the biomedical implementation of the PVA cryogels (artificial cartilages, in particular) are discussed in Section 3.3 devoted to the composite (filled) cryogels and cryostructurates.

### 4.4. Polymeric Cryogels and Cryostructurates as Technical Materials

According to the diagram in Figure 4, the cryogels and cryostructurates for the technical applications developed at the IOEC per se and also in collaboration with other studies are assigned as the materials for certain analytical systems, for the use as high-porous absorbents and filters, as well as for other engineering purposes. These areas include numerous individual “branches”, where the materials possessing particular specifics are necessary. Therefore, virtually in all such cases the individual R&D efforts were required for the elaboration of each particular system.

Rather evident examples, in which similar strictly “personal” approach was employed, are the so-called molecularly-imprinted polymeric matrices and the processes, mainly of analytical-type, where such materials were exploited [136,137,139,249]. Since the molecular imprinting, as a rule, tunes the polymeric matrix to the recognition of particular molecules, such systems in most cases are characterized by a high selectivity. In turn, if the imprinted material has wide interconnected pores, i.e., the matrix has a 3D morphology inherent in cryogels and cryostructurates, the binding of the specific ligands is fast, as it is not interfered by the diffusion barriers. For instance, such behavior was observed in the case of the sponge-like thermoresponsive poly(N-isopropylacrylamide-*co*-*N*-[3-(*N*,*N*’-dimethylamino)propyl]acryl-amide)-based cryogel, which has been synthesized in a frozen polymerizing system in the presence of ibuprofen as a template molecule (*59* in Table 1). The resultant imprinted cryogel possessed a molecular memory towards the template ligand: upon the thermo-induced collapse, the material has very quickly selectively bound the ibuprofen, but did not virtually bind its structural analogs such as benzoic, phenylacetic, and tolylacetic acids. Hereby, the combination of such polymeric matrix together with the DSC measurement of binding enthalpy gave the information on the binding efficiency, thus serving as an analytic technique.

The cryostructured wide-porous bioaffinity sorbents exhibiting directed specificity regarding the respective sorbates of a biological origin (enzymes, viruses, and whole cells) were also used for the analytical purposes. Thus, with the aid of the already mentioned pioneering metal-chelate affinity matrix carrying the Cu^2+^-loaded iminodiacetic acid groups (*35* in Table 2), it was possible within one population of *E. coli* cells to evaluate their amount, which had and no the available histidine units in proteins onto the cell surface. In another case, with the aid of the wide-porous agarose bioaffinity sorbents containing grafted aliphatic tails of different length (C_4_, C_7_, or C_12_) (*49* in Table 2), it turned out to be possible to analyze the amount of “young” and “old” cells of series of the bacterial strains after their culturing for different time periods. Thus, these examples demonstrate a number of valuable opportunities for the implementation of certain cryogenically-structured polymeric materials for solving various analytical tasks.

As for the absorbing and filtering cryogels and cryostructurates, such materials have also been prepared and their functionality has been examined. For instance, the dry polyelectrolyte-based cryostructurates consisting of the polymeric acids or bases (*26* and *27* in Table 1) were able to efficiently adsorb the vapors of the volatile alkaline and acidic substances, respectively. In addition, owing to their specific porosity, similar sponges (*24* in Table 2) have purified the passing air flow against the small dust particles, thus operating as the filter materials in the respective gas analyzer. The columnar reactor filled with the sponge-like cryogels on the basis of cross-linked chitosan derivatives (*52* in Table 2) absorbed the ions and retained the nano/microparticles of the radionuclides in the flow-through regime, i.e., purified the wastewater contaminated by these hazard pollutants. Other cryostructured matrices, namely the composite PVA cryogels filled with the small beads of the ion-exchange resins (*48b* in Table 1), selectively absorbed the counterions of the oppositely-charged solutes from the microbial cultural broths, but, owing to the impenetrable-for-cells shielding gel layers around the ion-exchanger, these composites did not bind the microbial cells resided in the same dispersion.

It has also been shown that the cryostructuring approach can be applied in many other technical fields. For instance, it was used for the processing of the tanned leather wastes by their transformation into, e.g., secondary leather (*16* in Table 2), or, more exactly, to the porous materials on the basis of tanned leather. Such manufactured articles, after the fat-liquoring and dyeing stages, could be used as thermo/sound-insulating decorative materials, the chemical composition of which is virtually the same as shoe leather.

Finally, the methods of the non-covalent cryostructuring that give rise to the formation of physical cryogels, the PVA-based ones primarily, are of significant value as the materials for the protection of ecology and in the construction engineering (see the Introduction) [5,16,148,149,150]. Researchers from the IOEC also participated in the early studies in these fields. In particular, it was related to the evaluation of the properties of the composite ices (*17* in Table 1) and to the elaboration of the technique for the fixation (reinforcement) of the defrosted grounds (*17* in Table 2). In the former case, a rather interesting effect was established: the addition of even a rather small amount of PVA in the composition of initial aqueous solution resulted in a marked decrease in the brittleness of the ice formed upon freezing of such solution. No doubts, this property is of significance for using the ices as the construction materials in the cold regions. In turn, the cryotropic gel-formation of the PVA solution injected into the grounds that are further subjected to the “natural cryogenic processing” was shown to allow protecting such grounds against the washing-out by the thawed waters. A similar technology is already applied in some cases for the preparation of anti-filtration screens and other watertight elements for dams of thawed or frozen types and reinforcement of different ground beds for roads and railways under the conditions of permanently and seasonally frozen soils [148,149]. When the ambient temperature increases to positive values during thaws or in spring, the frozen system is defrosted and forms a ground-filled PVA cryogel, the layer of which blocks the water filtration flows through the bodies of dams and prevents the penetration of the defrosted groundwater through the basements of the banquettes of the transport structures.

Certainly, the above described examples do not limit the possible technical applications of various cryogels and cryostructurates, including some of those developed at the IOEC over more than 40 years of R&D activity in the framework of this topic. The author of the present review believes that one can see a very wide applied potential of such polymeric materials as in the areas, where they are already implemented, as well as for the tasks that will arise in the near and distant future. Especially, as it is thought, the biomedical problems are the most probable ones.

## 5. Conclusions

Currently, the studies on cryostructuring processes and the development of related novel polymeric materials are intensively carried out in many countries. This assertion is illustrated by the origins of the respective review articles published by authors from all continents (excluding Antarctica, of course): Africa [71], Asia [8,10,11,14,18,20,24,25,30,36,37,39,42,45,48,49,53,62,69,70,73,78,80,83,84,102,106,108,117,122,137,138,139,142,143,145], Australia [59,119,126,127], Europe [6,7,19,21,22,27,28,29,32,33,34,35,40,41,43,44,47,50,51,55,56,57,58,61,65,75,76,81,93,95,97,98,99,104,105,110,114,115,118,121,123,128,129,131,133,134,135,136,147,148,149,150], North America [15,17,38,52,54,60,63,64,72,74,77,79,82,103,107,109,113,116,119,132,140,141], and South America [31,124]. With that, this list does not include the papers of the IOEC studies in order to demonstrate the actuality and importance of the given branch of polymer science not only for the author and his institution, but commonly all over the world. In addition, when the preparation of this paper was virtually finished, several new reviews were published. These articles are related to various aspects of the preparation and properties of the diverse polymeric cryogels and cryostructurates, as well as to their application in such fields as materials science [414,415,416], biomedical technologies [417,418,419,420,421], biotechnology [422,423], food processing [424], and ecological problems and construction engineering [425]. Thus, such permanent publishing activity once again testifies to the undiminishing interest for the cryostructuring topic.

## Figures and Tables

**Figure 1 gels-06-00029-f001:**
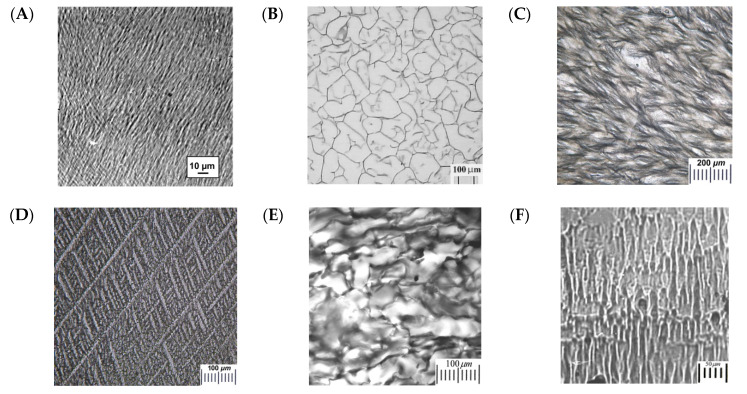
Micrographs demonstrating the macroporous morphology of various polymeric cryogels. and cryostructurates: (**A**) PVA-based cryogel (*51a* in Table 1) formed in a frozen (−20 °C/12 h) aqueous system (10-μm-thick thin section stained with Congo red; transmission optical microscope). (**B**) Agarose-based cryogel (*53a* in Table 1) formed in a frozen (−30 °C/1 h followed by −10 °C/23 h) aqueous system (10-μm-thick thin section stained with Cresyl violet; transmission optical microscope). (**C**) Polystyrene-based cryostructurate formed in the crystallized (+25 °C/4 h) naphthalene followed by its extraction with methanol (1-mm-thick layer; transmission optical microscope). (**D**) Butadiene-*co*-styrene-latex-based cryostructurate (*65* in Table 1) formed in a frozen (−20 °C/12 h) aqueous system (1-mm-thick layer; transmission optical microscope). (**E**) Poly(NIPAAM-co-DMAPA)-based cryogel (*59* in Table 1) formed in a frozen (−10 °C/20 h) aqueous system (10-μm-thick thin section stained with Bromophenol blue; transmission optical microscope). (**F**) Poly (acrylamide)-based cryogel (*5b* in Table 1) formed in a frozen (−8 °C/48 h) formamide using a unidirectional freezing (freeze-dried sample; transmission optical microscope). (**G**) Bovine-serum-albumin-based cryogel (*78b* in Table 1) formed in a frozen (−20 °C/18 h) aqueous system (1-mm-thick layer stained with Methylene blue; optical stereomicroscope). (**H**) Ca-Alginate-based cryostructurate (*76* in Table 1) formed in a frozen (−20 °C/1 h) aqueous system followed by freeze-drying and cross-linking with Ca-ions (1-mm-thick layer; transmission optical microscope). (**I**) Gelatin-based cryogel (*59c* in Table 2) formed in the medium of frozen DMSO (−20 °C/2 h) followed by its cryoextraction with cold (−20 °C) ethanol (2-mm-thick layer stained with Methylene blue; laser confocal microscope). Not that all these micrographs are from the private archive of the author of the present review and were taken by him personally.

**Figure 2 gels-06-00029-f002:**
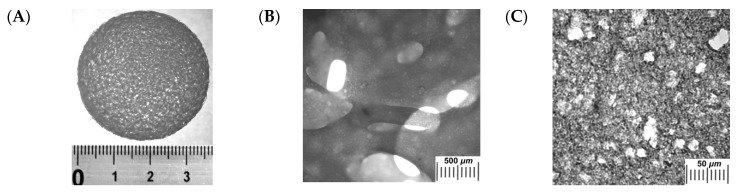
Macroporous morphology of the PVA cryogels prepared by the combination of the cryotropic. gel-formation and the additional phase-transformation processes: (**A**–**C**) Respectively, the appearance of the stained with Congo red 2-mm-thick disk (*68*, Table 1) prepared by freezing (−30 °C/0.5 h and then −5 °C/12 h) of the mixed solution containing PVA (7 wt.%) and gum arabic (7 wt.%) and the optical microscopy images of this gel material under different magnifications. (**D**–**F**) PVA-based cryogels (*72*, Table 1) formed by freezing (−11.6 °C/12 h) of the DMSO solutions of the polymer without (**D**) and with the additives of methanol in concentration of 1.70 mol/L (**E**) and 2.55 mol/L (**F**) (10-μm-thick thin section stained with Congo red; transmission optical microscope). (**G**–**I**) PVA-based cryogels (*62*, Table 1) formed by freezing (−25 °C/12 h) of the aqueous solutions of the polymer without (**G**) and with the additives of methanol in concentration of 1.23 mol/L (**H**) and 1.85 mol/L (**I**) (10-μm-thick thin section stained with Congo red; transmission optical microscope). Note that all these micrographs are from the private archive of the author of the present review and were taken by him personally.

**Figure 3 gels-06-00029-f003:**
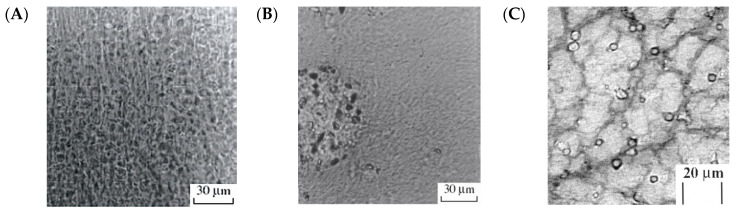
Micrographs demonstrating the influence of different dispersed fillers on the macroporous. morphology of various composite PVA cryogels (all the images were taken using transmission optical microscope for the 10-μm-thick thin sections stained with Congo red): (**A**) PVA-based composite cryogel (*45b*, Table 1) formed in a frozen (−20 °C/24 h) aqueous system containing suspended Silasorb−300 silica particles of 5 μm in size. (**B**) PVA-based composite cryogel (*45b*, Table 1) formed in a frozen (−20 °C/24 h) aqueous system containing suspended Silasorb-C18 hydophobized silica particles of 7.5 μm in size. (**C**) PVA-based composite cryogel (*50*, Table 1) formed in a frozen (−20 °C/12 h) aqueous system containing added tetramethoxysilane, hydrolytic polycondensation of which gave rise to the formation of silica filler particles. (**D**) PVA-based composite cryogel (*69b*, Table 1) formed in a frozen (−20 °C/12 h) aqueous system containing PVA and chitosan hydrochloride followed by the in situ transformation of the soluble chitosan salt to the particulate water-insoluble chitosan-base. (**E**) Microdroplets of the Vaseline oil mechanically dispersed in the aqueous PVA solution. (**F**) Composite PVA cryogel (*56a*, Table 1) formed on the basis of above (**E**) emulsion by its cryogenic processing (−20 °C/12 h). (**G**–**I**) Foamed PVA cryogels (*49*, Table 1) prepared by cryogenic processing (−30 °C/1 h and further −20 °C/23 h) of the following fluid foams: air-whipped aqueous PVA solution (**G**) foam stabilized with the additives of Brij−56 non-ionic surfactant; and (**H**) foam stabilized with the additives of CTAB cationic surfactant (**I**) Note that all these micrographs are from the private archive of the author of the present review and were taken by him personally. CTAB, cetyltrimethylammonium bromide; Brij−56, decaoxyethylene cetyl ether.

**Figure 4 gels-06-00029-f004:**
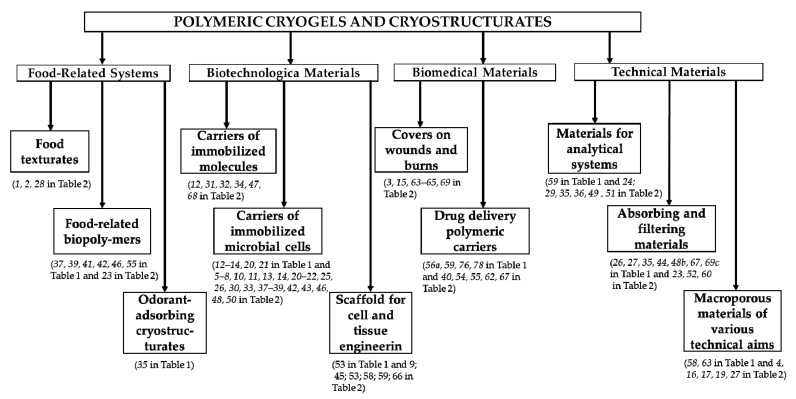
Schematic diagram showing the applied R&D activities of the research with respect of the developed in IOEC cryogenically-structured materials.

**Table 1 gels-06-00029-t001:** Basic research performed in the IOEC in the frameworks of the topics related to the cryogenic structuring processes.

Example	Precursors	Dispersion Medium	Conditions of Cryogenic Processing	What was Studied and Found	References	Publication Year
*1*	Thiol-containing poly(acylamide)	Aqueous medium	–8 °C/18 h	Detection that water-dissolved oxygen causes the formation of disulfide-crosslinked macroporous cryogels	[166]	1981
*2*	The systems: polymer + cross-linker, vinyl and divinyl comonomers + initiator; polycondesation comonomers	Aqueous and organic solvents	1–60 °C below the solvent crystallization point/1–72 h	Elaboration of the synthetic approaches to the preparation of the chemically cross-linked cryogels	[167]	1982
*3*	The systems: polymer + cross-linker	Water; DMSO	–8 °C/18 h; +2 °C/15 h	Demonstration of the possibility to prepare chemically crosslinked cryogels both in aqueous and organic non-deeply-frozen media	[168]	1982
*4a* *b* *c*	Chitosan + glutar aldehyde	Weakly acidic aqueous medium	−8 to −30 °C/18–24 h	Detection of the effect of an apparent decrease in the critical concentration of the cryotropic gelation	[169] [170] [171]	1982 2010 2011
*5a* *b*	AAM + MBAAM + TEMED + APS	Water FA	−8 °C/18 h −10 °C/24 h	Preparation of poly(acrylamide)-based polymerization-type cryogels in frozen aqueous or formamide media and the studies of the resultant cryogels properties	[172] [173,174]	1983 1984
*6a* *b* *c* *d* *e*	Gelatine; Agar-agar; Agaroide; PVA	Water DMSO	−10 °C/24 h −10 to −20 °C/19–24 h −10 °C/24 h	Demonstration of general character of the phenomenon of the formation of non-covalent (physical) cryogels	[175]	1984
*7a* *b*	Poly(NVP-co-MA) + DADPO; PS + PXDC + SnCl_4_	DMSO NB	−8 °C/18 h −4.5 to −27 °C/1–24 h	Preparation of covalent cryogels in frozen organic media and studies of the resultant cryogels properties; the first use of the term “cryogels”	[176]	1984
*8*	AAM + MBAAM + TEMED + APS	Water	−10 to −30 °C/1–48 h	Detection of the gel-formation acceleration effects and of a bell-shaped dependence of the gel-formation efficiency on the cryogenic processing temperature	[177]	1986
*9*	PVA	Water	−10 to −20 °C/12–240 h	The evidence of the non-covalent nature of the intermolecular links in the PVA cryogels	[178]	1986
*10*	PVA	Water	−10 to −20 °C/1–240 h	Studies of the influence of PVA molecular weight and concentration; freezing temperature and duration on the physico-mechanical properties of the PVA cryogels and their microstructure	[179]	1986
*11*	PVA	Water	−10 to −30 °C/1–24 h	The first demonstration of the principal significance of the heating rate upon the frozen samples defrosting for the properties of the resultant PVA cryogels	[180]	1988
*12*	PVA + *Aspergillus clavatus* spores	Aqueous medium	−20 °C/20 h	Studies of the biological activity of immobilized biocatalyst capable of producing the guanyl-ribonuclease	[181]	1988
*13*	AAM + MBAAM + TEMED + APS + *Escherichia coli* cells	Aqueous medium	−10 to −30 °C/18–24 h	Studies of the influence of cryopolymerization conditions on the microstructure of cryogel-based carriers, as well as on the cells viability and activity	[182]	1988
*14*	Sodium silicate + HCl + *Renobacter vacuolatum* (or *Blastobacter viscosus*, or *Methhylobacterium*) cells	Aqueous media, pH ~6	−10 °C/24 h	Preparation of immobilized biocatalysts (bacterial cells entrapped in cryo-silicagel) exhibiting the hydrogen-oxidizing activity and studies of the physico-chemical properties and activity of such biocatalytic systems	[183]	1988
*15*	AAM + MBAAM + TEMED + APS	Water	−10 to −30 °C/24 h or −196 °C/0.5 h followed by −10 to −30 °C/24 h (low-temperature quenching technique)	Studies of the dynamics of cryotropic gel-formation, evaluations of the properties and microstructure of the resultant gel matrices as dependent on the freezing techniques	[184]	1989
*16a* *b*	Thiol-containing poly(acrylamide) derivative	Aqueous medium, pH 7.5	−10 to −30 °C/1–24 h	Studies of the dynamics of cryotropic gel-formation and the measurements of the amount of reactive pendant SH-groups of the polymer during such process	[185] [186]	1989 2000
*17*	PVA	Water	−10 to −30 °C/24–240 h	Studies of the properties of the PVA cryogels as dependent on the frozen storing duration	[187,188]	1989
*18*	PVA	Water	−10 to −20 °C/1–24 h	Studies of the thermomechanical properties of the PVA cryogels formed under various thermal regimes	[189]	1989
*19*	PVA	D_2_ O	−20 to −66 °C/1–6 h	^2^H and ^13^C NMR studies of the frozen PVA solutions	[190]	1990
*20*	PVA + yeast cells	Aqueous medium	−15 °C/19 h	Studies of the filler particles (yeasts) influence on the physico-mechanical properties of the resultant composite PVA cryogels	[191]	1990
*21*	AAM + MBAAM + TEMED + APS + cells (*Renobacter vacuolatum,* or *Blastobacter viscusus*, or *Methhylobacterium* sp.)	Aqueous medium	−10 °C/24 h	Preparation of immobilized biocatalysts; studies of their physico-chemical properties and hydrogen-oxidizing activity	[192]	1991
*22*	PVA + nitroxyl-type spin probe	Water	−5 to −20 °C during the time of measurements	The ESR spectroscopic measurements of the UFLMP volume at various minus temperatures	[193]	1991
*23a* *b*	PVA + oligoethylene glycols	Water	−5 to −20 °C/1–240 h −20 °C/18 h	Preparation and studies of the complex PVA cryogels that contained the additives of various oligoethylene glycols	[194] [195]	1992 1995
*24*	PVA + particles of cross-linked dextran (series of Sephadexes)	Water	−15 °C/18 h	Detection of the key role of the filler porosity for its influence on the properties of the composite PVA cryogels	[196]	1992
*25*	AAM + MBAAM + TEMED + APS	D_2_O	−13 to −28 °C/1–60 min or −196 °C/0.5 h followed by −13 to −28 °C/1–60 min (low-temperature quenching technique)	^1^H- and ^2^H NMR studies of the poly(acrylamide) cryogels formation in frozen aqueous systems	[197]	1993
*26*	Water-soluble salts of polyacids	Aqueous media	−15 to −78 °C/1–12 h	Approaches for the preparation of the polyacids-based spongy cryostructurates	[198]	1994
*27*	Water-soluble salts of polymeric bases	Aqueous media	−8 to −78 °C/2–24 h	Approaches for the preparation of spongy cryostructurates consisting of the polymeric bases	[199]	1994
*28*	PVA + low-molecular polyols	Water	−20 °C/18 h	Preparation and studies of the PVA cryogels that contained the additives of low-molecular polyols	[195]	1995
*29a* *b*	HEMA + MBAAM + TEMED + SPS	Water Water + dioxane	−8 to −10 °C/120 h −8 to −18 °C/24–75 h	Synthesis and studies of the poly(hydroxyethyl methacrylate)-based cryogels prepared in frozen aqueous of mixed water-organic media	[200] [201]	1996 1996
*30*	PVA	Water	−20 °C/24 h	Studies of the PVA cryogels swelling behavior in the salt-containing aqueous solutions	[202]	1996
*31*	PVA + low-molecular electrolytes	Water	−20 °C/24 h	Studies of the low-molecular electrolytes influence on the formation and properties of the PVA cryogels	[203]	1996
*32*	DEAAM + MBAAM + TEMED + APS	Water	−24 °C/0.5 h and further −10 °C/18 h	Preparation and studies of the first example of the temperature-responsive polymeric cryogels	[204]	1997
*33a* *b*	PVA + disperse mineral or organic fillers of various porosity	Water	−15 °C/18 h −10 to −30 °C/12 h	Studies of the influence of micro-, meso- and macroporous fillers on the properties of the composite PVA cryogels	[205] [206]	1997 2017
*34*	Ovalbumin + urea	Water	−8 to −32 °C/0.1–48 h	Studies of the denaturation-induced cryotropic gel-formation of ovalbumin	[207]	1997
*35a* *b* *c* *d*	Gelatinized starch + aroma compounds	Aqueous media	−18 °C/16 h	Preparation of the starch-based cryogels that contained various aroma compounds and study of their adsorption and release	[208] [209] [210,211] [212,213]	1998 1999 2001 2002
*36*	PVA	Water	−10 to −30 °C/24 h	Studies of the cryostructuring processes in the low-concentrated PVA solutions	[214]	1999
*37*	Microparticulate dispersions of fibrous collagen	Aqueous media, pH 1.5–12.4	−15 °C/16–18 h	Studies of the microfibrillar collagen based cryogels that were formed at various pH values of the feed systems	[215]	2000
*38*	PVA	Water	−18 °C/18 h	Evaluation of the apparent yield of the PVA cryotropic gel-formation	[216]	2000
*39*	Amylopectin	Water	−6 to −24 °C/18 h	Studies of the cryoprecipitation effects in the low-concentrated solutions of amylopectin	[217]	2000
*40*	PVA	Water	−20 °C/24 h	Studies of the PVA cryotropic gel-formation dynamics	[218]	2000
*41*	Amylopectin + amylose + NaCl	Water	−6 to −24 °C/18 h	Studies of the cryostructuring processes in frozen aqueous solutions of the amylopectin/amylose mixtures	[219]	2000
*42*	Locust bean gum	Water	−20 °C/18 h	Studies of the locust bean gum cryotropic gel-formation and the elucidation of the nature of intermolecular links in the resultant cryogels	[220]	2000
*43*	PVA + gas bubbles	Water	−10 to −30 °C/18 h	Preparation and studies of the gas-filled (foamed) PVA cryogels	[221]	2001
*44*	Agarose + gelation-slowing additives	Aqueous media	−5 to −50 °C/1–24 h	Preparation and studies of the agarose-based wide-pore cryogels	[222,223]	2001
*45a* *b*	PVA + hydophobic disperse particles	DMSO	−15 to −30 °C/1–24 h	Development of the approach for the preparation of composite PVA cryogels with entrapped hydrophobic fillers; studies of the properties of such composites	[224] [225]	2001 2002
*46*	Maltodextrin	Water	−6 to −24 °C/18 h	Studies of cryostructuring of the maltodextrin-containing aqueous solutions	[226]	2002
*47*	PVA + surfactants	Water	−5 to −78 °C/5–48 h	Revelation of the possibility to affect the macroporous morphology of PVA cryogels with the aid of surfactant additives	[227]	2003
*48a* *b*	PVA + particles of ion-exchange resins	Water	−20 °C/5.5 h	Studies of the charged fillers influence of the properties and microstructure of composite PVA cryogels	[228] [229]	2004 2005
*49*	PVA + gas bubbles + surfactants	Water	−20 to −30^o^C/24 h	Studies of the surfactants influence on the physico-chemical properties and porous structure of the gas-filled (foamed) PVA cryogels	[230]	2005
*50*	PVA + TMOS	Water	−15 to −30 °C/12 h	Preparation and studies of the hybrid organic–inorganic PVA cryogels formed with simultaneous sol–gel transformation of TMOS precursor	[231]	2007
*51a* *b* *c*	PVA	Water	−10 to −40 °C/1–240 h	Systematic studies of the influence of all parameters of the cryostructuring processes on the properties and structure of PVA cryogels	[232] [233] [234]	2007 2008 2012
*52a* *b*	Poly(acrylamide) + glutar aldehyde	Aqueous medium, pH 10	−5 to −20 °C/24 h	Preparation and studies of the covalently-linked macroporous cryogels based on pre-synthesized poly(acrylamide)	[235] [236]	2007 2008
*53a* *b* *c* *d*	Agarose + gelation-slowing additives	Aqueous media	−5 to −30 °C/24 h	Preparation of the wide-pore agarose cryogel with grafted gelatin and its use as a scaffold for culturing of insulin-producing cells	[237] [238] [239] [240]	2008 2005 2007 2010
*54a* *b*	NIPAAM + MBAAM + TEMED + APS + oil microdroplets	Aqueous medium	−15 °C/24 h	Preparation and studies of composite thermoresponsive wide-pore cryogels filled with oil dispersion	[241] [242]	2008 2013
*55*	Gelatinized starch + gluten or gums additives	Aqueous medium	−9 to −40^o^C/23 h	Studies of the influence of gluten and gums (guar, xanthan) additives on the physico-chemical properties and macro-porous morphology of the complex starch-based cryogels	[243]	2009
*56a* *b*	PVA + microdroplets of hydrophobic liquids	Aqueous media	−10 to −60 °C/1–24 h	Studies of composite PVA cryogels filled with microdroplets of hydrophobic liquids	[244] [245]	2010 2012
*57*	PVA + alkali metal chlorides	Water	−20 °C/12 h	Studies of the influence of alkali metal chloride additives on the formation, properties and macroporous morphology of the PVA cryogels	[246]	2011
*58*	DMAAM + ionic acrylates + TEMED + APS	Water	−5 to −40 °C/4–24 h	Method for the preparation of poly(*N*,*N*’-dimethylacrylamide)-based cryogels that possess the superabsorbent properties and study of the mechanisms for such materials formation	[247] [248]	2011 2014
*59*	NIPAAM + DMAPA + BAC + TEMED + APS + ibuprofen	Aqueous medium	−10 °C/20 h	Synthesis of the ibuprofen-templated wide-pore thermoresponsive cryogels and the studies of their specificity with respect of various ligands	[249]	2011
*60*	PVA + dispersed PVAc microparticles	Aqueous medium	−20 °C/12 h	Preparation of composite PVA cryogels and studies of the influence of particulate polymeric fillers on the properties and microstructure of the resultant gel materials	[250]	2012
*61*	PVA + guar gum	Aqueous media, pH 5–11	−5 to −30 °C/12 h	Preparation and studies of the wide pore PVA cryogels using a combination of the liquid–liquid phase separation and the cryotropic gel-formation processes	[251]	2012
*62*	PVA + series of the low-molecular aliphatic alcohols	Aqueous medium	−15 to −35 °C/12 h	Studies of the influence which the additives of low-molecular aliphatic alcohols exert on the physico-chemical properties and macroporous morphology of the PVA cryogels	[252]	2014
*63*	PVA	Water	−10 to −50 °C/1–48 h	Elaboration of the approach for the secondary molding of the PVA cryogels	[253]	2014
*64*	PVA + dispersions of butadiene-co-styrene latex	Water	−20 °C/12 h	Preparation and studies of the latex-filled composite PVA cryogels	[254]	2015
*65*	Aqueous dispersions of butadiene-*co*-styrene latex	Water	−20 °C/12 h	Preparation of the latex-based cryostructurates and studies of their microstructure	[254]	2015
*66*	DMAAM + allyl derivatives of 1,8-naphthalimide + TEMED + APS	Water	−10 to −30 °C/24 h	Preparation and studies of the osmotic and optical properties of fluorescent copolymeric cryogels	[255]	2015
*67*	PVA + chitosan powder	Water	−20 °C/12 h	Preparation of chitosan-filled composite PVA cryogels, studies of their properties, microstructure and absorption capacity with respect of copper ions	[256]	2015
*68*	Serum albumin + urea + cystein	Aqueous medium	−15 to −25 °C/20 h	Preparation of the wide-pore serum-albumin-based cryogels; studies of their physico-chemical properties, porosity characteristics and the mechanisms of the 3D network formation	[257]	2015
*69a* *b* *c* *d*	PVA + chitosan hydrochloride	Aqueous medium	−20 °C/12 h	Preparation of complex and composite chitosan-containing PVA cryogels with the use the in situ filler formation, studies of their properties, microstructure and absorption characteristics	[258] [259] [260] [261]	2016 2017 2019 2020
*70*	Serum albumin + EDC	Aqueous medium	−15 to −25 °C/20 h	Preparation and studies of EDC-assisted cross-linked albumin cryogels, studies of their properties and macroporous morphology	[262]	2016
*71*	Serum albumin	Water	−15 to −25 °C/18 h	Preparation of the wide-pore albumin-based cryostructurates followed by their chemical tanning and studies of the properties	[263]	2017
*72*	PVA + low-molecular aliphatic alcohols	DMSO	−11.6 to −31.6 °C/12 h	Studies of the influence of low-molecular aliphatic alcohol additive on the properties and microstructure of the PVA cryogels formed in a frozen DMSO medium	[264]	2017
*73a* *b*	PVA + organic chaotropes and kosmotropes	Aqueous media	−20 °C/12 h	Studies of the influence of chaotropic and kosmotropic additives on the physico-chemical properties and microstructure of PVA cryogels	[265] [266]	2018 2019
*74*	PVA + organic chaotropes and kosmotropes	DMSO	−11.6 to −31.6 °C/12 h	Studies of the influence of chaotropic and kosmotropic additives on the physico-chemical properties and microstructure of the PVA cryogels formed in a frozen DMSO medium; revealing of the unexpected “kosmotropic-like” effects of the organic chaotropes	[267,268]	2018
*75*	PVA + cellulose-based fillers + salting-out electrolyte	Aqueous medium	−20 °C/12 h	Studies of the combined fillers and electrolytes influence on the properties and structure of composite PVA cryogels	[269]	2019
*76*	Sodium alginate	Water	−10 to −30 °C/1 h	Preparation of Ca-alginate-based wide-pore cryostructurates, studies of their properties and porous morphology, as well as the evaluation of their suitability as drug delivery carriers	[270]	2019
*77*	PVA + TMOS + aqueous HCl as a catalyst	DMSO	−21.6 °C/18 h	Preparation and studies of the hybrid organic–inorganic PVA cryogels formed in the frozen DMSO medium	[271]	2019
*78a* *b*	BSA + urea + cysteinBSA + GHC + cystein	Aqueous media, pH ~8	−15 to −25 °C/18 h	Preparation of the wide-pore albumin-based cryogels followed by their chemical modification via the succinylation, studies of the properties of the resultant spongy matrices and evaluation of their potential as drug delivery materials	[272]	2020

Abbreviations: AAM, acrylamide; AGE, allyl glycidyl ether; APS, ammonium persulphate; BAC, *N*,*N*’-bis(acryloyl)cystamine; BSA, bovine serum albumin; DADPO, 4,4′-diaminodiphenyloxide; DEAAM, *N*,*N*’-diethylacrylamide; DMAAM, *N*,*N*’-dimethylacrylamide; DMAPA, *N*-[3-(*N*,*N*’-dimethylamino)propyl]-acrylamide; DMSO, dimethylsulfoxide; FA, formamide; GHC, guanidine hydrochloride; HEMA, hydroxyethyl methacrylate; MA, maleic anhydride; MBAAM, *N*,*N*’methylene-bis-acrylamide; NB, nitrobenzene; NIPAAM, *N*-iso-propylacrylamide; NVP, *N*-vinylpyrrolidone; PS, polystyrene; PVA, poly(vinyl alcohol); PVAc, poly(vinyl acetate); PXDC, p-xylylene dichloride; SPS, sodium persulphate; TMOS, tetramethoxysilane.

**Table 2 gels-06-00029-t002:** The examples of the R&D studies directed to the applied elaborations that were and are now performed in the IOEC.

Example	Precursors	Dispersion Medium	Conditions of Cryogenic Processing	Elaboration Targets and Related Studies	References	Publication Year
*1*	Coagulates of the plant proteins isolates	Water	−10 to −60 °C/1–48 h	Method for the preparation of the porous gels based on plant proteins	[163]	1972
*2a* *b* *c*	Paste-like dispersion of fish or crustacean proteins isolates	Aqueous media, pH 5–7	−13 to −15 °C/1.5–48 h	Method for the preparation of the cryogels based on myofibrillar proteins; studies of the cryogels properties	[164] [273] [274]	1977 1981 1984
*3*	Sodium alginate	Water	−20 to −40 °C/0.5–24 h	Elaboration of the Ca-alginate cryostructurates that are of interest as the biomedical materials	[275,276]	1985
*4*	PVA + nutrients	Water	−5 to −196 °C/0.5–24 h	Elaboration of the PVA-cryogel-based microbiological solid nutritional media	[277]	1985
*5*	PVA + microbial cells	Aqueous media	−10 to −70 °C/1–240 h	Method for the entrapment of microbial cells in the PVA cryogel carriers	[278]	1986
*6*	PVA + *Citrobacter intermedius* cells	Aqueous medium	−10 to −30 °C/12–24 h	Development and studies of the immobilized biocatalysts for the production of 3-fluoro-L-tyrosine	[279] [280]	1986 1989
*7*	AAM + MBAAM + TEMED + APS + *Saccharomyces cerevisiae* cells	Aqueous medium	−10 to −30 °C/18–36 h	Immobilized biocatalysts capable of ethanol producing	[281]	1986
*8*	PVA + spores of filamentous fungi	Aqueous medium	−10 to −40 °C/8–24 h	Immobilized biocatalysts capable of ribonucleases producing	[282]	1986
*9*	PVA + nutrients	Aqueous media	−18 °C/3 h	Solid nutritional media for the cultivation of plant tissues and cells	[283]	1986
*10*	PVA + *Saccharomyces cerevisiae* cells	Aqueous medium	−70 °C/5–10 min	Immobilized biocatalysts capable of acidic phosphatase producing	[284]	1986
*11*	PVA + *Zymomonas mobilis* cells	Aqueous medium	−10 to −60 °C/12–24 h	Immobilized biocatalyst for the production of ethanol	[285] [286]	1986 1996
*12*	AAM + MBAAM + AGE + TEMED + APS	Water	−20 °C/24 h	Elaboration of the cryogel-based carriers for the covalent immobilization of serum albumin	[287]	1987
*13*	PVA + *Erwinia aroidea* cells	Aqueous medium	−6 to −30 °C/12–48 h	Immobilized biocatalysts for the biotransformation of fumaric acid to L-aspartic acid	[288]	1988
*14*	PVA + *Alcaligenes faecalis* cells	Aqueous medium	−6 to −30 °C/12–48 h	Immobilized biocatalysts for the decarboxylation of L-aspartic acid to L- alanine	[289] [290]	1988 1990
*15a* *b* *c*	Dispersion of collagen coagulate + gluar aldehyde	Aqueous medium	−5 to −20 °C/10–15 h	Methods for the preparation of collagen sponges	[291] [292] [293]	1988 1992 1995
*16a* *b* *c* *d*	Dispersion of tanned leather particles + gluar aldehyde	Aqueous media, pH 3.5–5.5	−5 to −60 °C/1–24 h	Methods for the preparation of leather-like materials based on the tanned leather wastes	[294,295] [296] [297] [298]	1988 1989 1992 1995
*17*	PVA + dispersed ground particles	Aqueous media	−2 to −60 °C/3–24 h	Method for the fixation of thawed grounds	[299]	1990
*18a* *b*	PVA	Water	−5 to −40 °C/1–24 h	The variants of setup for the preparation of the beaded PVA cryogels	[300] [301]	1992 1996
*19*	PVA	Water	−1 to −60 °C/1–12 h	Method for the preparation of the PVA-cryogel-based beaded artificial baits for sporting and amateur fishing.	[302]	1992
*20*	PVA + *Presudomonas* sp. Cells	Aqueous medium	−20 °C/24 h	Preparation and study of immobilized biocatalyst for the use in a biosensor for L-proline detection and quantification	[303]	1992
*21*	PVA + *Corynebacterium glutamicum* cells	Aqueous medium	−20 °C/15 h	Immobilized biocatalysts for the biosynthesis of L-lysine	[304]	1992
*22*	PVA + *Clostridium thermosaccharolyticum* cells	Aqueous medium	−20 °C/24 h	Thermotolerant immobilized biocatalyst capable of hydrogen producing	[305]	1993
*23*	Water-soluble proteins + denaturants	Aqueous media	−3 to −196 °C/0.2–48 h	Methods for the preparation of macroporous proteinaceous cryogels	[306]	1994
*24*	Water-soluble salts of polyacids	Aqueous media	−15 to −78 °C/1–12 h	Application of the polyacids-based spongy cryostructurates in analytical systems	[307]	1994
*25a* *b*	PVA + thermotolerant *Acetogenium kivuii* cells	Aqueous medium	−20 °C/24 h	Preparation and study of the immobilized biocatalyst for the production of acetate via the CO_2_ reduction	[308,309] [310]	1994 1996
*26a* *b*	PVA + *Arthrobacter globiformis* cells	Aqueous media	−20 °C/24 h	Preparation and study of the immobilized biocatalysts for the biotransformation of sterol compounds	[311] [312]	1995 1996
*27a* *b*	Latex-containing dispersions	Aqueous media	−4 to −100 °C/0.5–12 h	Method for the preparation of the latex-based porous cryogenically structured materials	[313] [314,315,316]	1996 1998
*28*	Dispersed particles of fibrillar proteins	Aqueous media	−5 to −40 °C/6–12 h	Method for the preparation of macroporous cryogels based on fibrillar proteins	[317]	1997
*29a* *b*	PVA	Water	−20 °C/24 h	Covalent immobilization of specific antibodies in the beaded PVA cryogels and the use of such macroporous immunosorbents for the isolation of viruses	[318] [319]	1997 1998
*30*	PVA + *Kluyveromyces marxianus* cells	Aqueous medium	−20 °C/18 h	Preparation and study of the immobilized biocatalyst with the PVA-cryogel-entrapped thermotolerant yeasts for the production of ethanol at elevated temperatures	[320]	1998
*31* *a* *b* *c* *d* *e* *f* *g* *h*	PVA	Water	−20 °C/24 h	Preparation of the beaded PVA cryogel, subsequent covalent immobilization of enzymes and the use of the resultant biocatalysts for the operation in organic media. The enzymes: -α-chymotrypsin -hog pancreas lipase -subtilisin 72 -thermolysin	[321] [322] [323] [322] [324,325] [326,327] [328,329] [325]	2000 2005 2000 2005 2005 2001 2008 2008
*31i* *j* *k*				-thermolysin -trypsin -subtilisin Carlsberg	[330] [331] [332]	2008 2001 2003
*32a* *b*	PVA + dispersions of the cross-linked enzyme aggregates	Aqueous medium	−5 to −40 °C/4–48 h	Method for the preparation of the composite immobilized biocatalysts filled with disperse particles of the cross-linked enzyme aggregates	[333] [334]	2002 2009
*33a* *b*	PVA + cells of lactic-acid-producing microorganisms	Aqueous media	−15 to −30 °C/6–30 h	Methods for the preparation of the immobilized biocatalysts capable of producing lactic acid and the studies of such biocatalysts	[335] [336,337]	2002 2006
*34*	PVA	Water	−20 °C/24 h	Preparation of immobilized OPH using additives of Polybrene as a protective agent for the enzyme activity	[338]	2002
*35*	AAM + AGE + MBAAM + TEMED + APS	Aqueous medium	−12 to −15 °C/5–15 h	Synthesis and study of the supermacroporous affinity and ion exchange columns for the isolation of microbial cells	[339]	2002
*36*	AAM + AGE + MBAAM + TEMED + APS	Aqueous medium	−10 °C/24 h	Synthesis and study of the supermacroporous metal-chelate affinity cryogels for the isolation and purification of enzymes	[340]	2003
*37a* *b*	PVA + Champagne yeast cells	Aqueous medium	−20 °C/24 h	Preparation and study of the immobilized biocatalysts for the fermentation of sparkling wines	[341] [342]	2003 2004
*38*	PVA + *Rhizopus oryzae* spores	Aqueous medium	−10 to −30 °C/12–24 h	Preparation and studies of the immobilized biocatalysts capable of producing L(+)-lactic acid	[343]	2004
*39*	PVA + suspension of soil microorganisms	Aqueous medium	−20 °C/18 h	Preparation and study of immobilized biocatalysts for the bioremediation of diesel-contaminated soils	[344]	2004
*40a* *b* *c*	Water-soluble neutral polysaccharides + dinitrosyl iron complexes	Aqueous media	−10 to −196 °C/0.1–24 h	Method for the preparation of the cryostructurates serving as the carries for the dinitrosyl iron complexes and the studies of these drug-release systems as the NO-donors	[345] [346] [347]	2005 2007 2008
*41*	PVA + microdroplets of hydrophobic liquids	Aqueous media	−10 to −60 °C/1–24 h	Method for the preparation of the composite PVA cryogels filled with microdroplets of hydrophobic liquids	[348]	2006
*42*	PVA + *Rhodococcus ruber* cells	Aqueous media	−18 °C/12 h	Preparation and studies of the immobilized biocatalysts for the oxidation of hydrocarbon pollutants	[349] [350]	2006 2009
*43a* *c* *c*	PVA + *Rhizopus oryzae* spores	Aqueous medium, pH 6.0	−16 to −22 °C/16–48 h	Preparation and studies of the immobilized biocatalysts for the bioremediation of industrial waters contaminated with food wastes	[351] [352] [96,353]	2006 2007 2008
*44*	Agarose + gelation-slowing additives	Aqueous medium	−5 to −20 °C/24 h	Preparation of the wide-pore agarose cryogel with grafted gelatin and its use as a scaffold for culturing of stem cells	[354,355]	2008
*45*	Series of synthetic and natural polymers	Aqueous media	−10 to −60 °C/1–48 h	Methods for the preparation of the cryogel-based carriers, their use as the wide-pore scaffolds for culturing of permissive cells followed by virus infection and propagation	[356]	2008
*46*	PVA + spores or cells of the pectinase-producing strains	Aqueous media	−13 to −20 °C/16–24 h	Preparation of the immobilized biocatalysts for the biosynthesis of pectinases	[357]	2008
*47a* *b* *c* *d*	PVA	Water	−20 °C/24 h	Preparation of the beaded PVA cryogel, subsequent covalent immobilization of laccase and the use of the resultant biocatalysts for the remediation of dyed textile waste waters	[358] [359] [360] [361]	2009 2010 2011 2012
*48*	AAM + MBAAM + TEMED + APS	Water	−12 °C/18 h	Preparation of the wide-pore poly(acrylamide) cryogels, their hydrophobization and the use for the adsorption immobilization of the *Rhodococcus* cells	[362] [363]	2009 2011
*49*	Agarose + gelation-slowing additives	Aqueous medium	−5 to −20 °C/24 h	Preparation of the wide-pore agarose cryogels, their hydrophobization and the use as the bioaffinity adsorbents for the separation of microbial cells	[364]	2009
*50*	PVA + OPH-bearing microbial cells	Aqueous medium	−20 °C/24 h	Preparation of the immobilized biocatalyst for the decomposition of the organophosphate toxins	[365]	2009
*51*	PVA + guar gum	Aqueous medium	−5 to −30 °C/12 h	Preparation of the wide-pore PVA cryogels, their hydrophobization and use as the bioaffinity adsorbents for the selective attachment of microbial cells	[366]	2010
*52*	Chitosan + glutar aldehyde	Aqueous medium, pH 5	−20 °C/24 h	Preparation of the wide-pore chitosan-based cryogel and its use for the sorption of the heavy metal ions	[367,368]	2011
*53*	Sodium alginate	Aqueous medium	−20 °C/12 h	Preparation of the Ca-alginate-based wide-pore cryostructurate, its chemical modification followed by the gelatin grafting and the subsequent use as scaffold for culturing and cryopreservation of stem cells	[369,370] [371] [372]	2011 2013 2014
*54*	Water-soluble biomedical polymers + hydrogenated pyrido(4,3-b)indole	Aqueous medium	−40 °C/1–24 h	Method for the preparation of the drug-containing cryostructurates for the treatment of neurodegenerative diseases	[373]	2011
*55a* *b*	Starch + lacto- and bifidobacterial cells	Aqueous media	−10 to −196 °C/0.1–6 h	Method for the preparation of the starch cryostructurates that contain probiotic bacteria; the studies of the formulation’s biological activity	[374] [375]	2012 2015
*56*	PVA + spores of various filamentous fungi	Aqueous medium	−20 °C/24 h	Preparation and studies of the immobilized biocatalysts capable of producing cellulases	[376]	2013
*57a* *b*	PVA + *Pimelobacter simplex* cells	Aqueous medium	−15 °C/18 h	Immobilized biocatalyst for the microbial biotransformation of sterols	[377] [378] [379]	2011 2013 2015
*58*	Sodium alginate	Aqueous medium	−20 °C/12 h	Preparation of Ca-alginate-based wide-pore cryostructurate, its chemical modification followed by the gelatin grafting and the use as scaffold for culturing, cryopreservation and in vivo transplantation of fetal liver cells	[380]	2015
*59a* *b* *c* *d*	Gelatin	DMSO	+3 to −22 °C/1–24 h	Elaboration of the method for the preparation of the gelatin-based cryostructurates under sterilizing conditions and the use of the resultant materials as wide-pore 3D scaffolds for culturing of human or animal cells	[381] [382] [383] [384]	2015 2016 2018 2019
*60*	Water-soluble polyelectrolyte + particles of the hypercrosslinked PS	Aqueous media	−10 to −40 °C/1–8 h	Method for the preparation of the composite wide-pore cryostructurates and their use as the sorbents for toxicants and pollutants	[385]	2015
*61*	Proteins of blood serum + urea + cystein	Aqueous media	−10 to −30 °C/6–36 h	Method for the preparation of the sponges based on proteins of blood serum, their loading with antibiotics and the use as antimicrobial biodegradable materials for the treatment of the infected wounds and burns	[386]	2016
*62*	PVA + series of amino acids	Aqueous media	−20 °C/24 h	Preparation of the PVA-cryogel-based carriers of biologically active substances	[387]	2017
*63a* *b*	Sodium alginate	Water	−20 °C/12 h	Preparation of the Ca-alginate-based wide-pore cryostructurates, their loading with nanocomposites dioxidine-metal (Ar, Cu) and the use as potential antibacterial materials	[388] [389]	2017 2018
*64*	Proteins of blood serum + urea + cystein	Aqueous medium	−20 °C/24 h	Preparation and in vivo studies of the proteinaceous sponges for the use as biocompatible coatings to restore the full-thickness excision wounds	[390]	2018
*65*	Gelatin	DMSO	−12 °C/24 h	Preparation of the gelatin-based wide-pore cryostructurates, their loading with nanocomposites dioxidine/metal (Ar, Cu) and the use as potential antibacterial materials	[389,391]	2018
*66*	Chitosan acetate	Aqueous media	−10 °C/15 h	Preparation of the chitosan-based cryostructurates followed by their tanning with genipin and the use as the wide-pore 3D-scaffolds for the culture of fibroblasts	[392]	2018
*67*	Chitosan; alginate; gelatin	Aqueous media	−15 to −30 °C/12–24 h	Preparation of the wide-pore biopolymeric cryostructurates and their use as the carriers of the osteogenic bioregulator	[393]	2019
*68*	PVA + bacterial nanocellulose + OPH	Aqueous medium	−20 °C/12 h	Preparation of the composite PVA cryogels possessing enzymatic activity and their use as potential antimicrobial biomaterials	[394]	2019
*69*	Chitosan; alginate; gelatin	Aqueous media	−15 to −30 °C/12–24 h	Preparation of biopolymeric wide-pore cryostruucturates, their loading with nanocomposites dioxidine/metal (Ar, Cu) or gentamicin/metal (Ar, Cu) and the use as potential antibacterial materials	[395]	2020
*70*	PVA + antibiotics	Aqueous media		Temporary implants for treating of infected bone defects	[396]	2020

Abbreviations: AAM, acrylamide; AGE, allyl glycidyl ether; APS, ammonium persulphate; DMSO, dimethylsulfoxide; MBAAM, *N*,*N*’methylene-bis-acrylamide; OPH, organophosphate hydrolase; PVA, poly(vinyl alcohol).

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
