# Peer review of "Cryostructuring of Polymeric Systems. 55. Retrospective View on the More than 40 Years of Studies Performed in the A.N.Nesmeyanov Institute of Organoelement Compounds with Respect of the Cryostructuring Processes in Polymeric Systems"

_gels, 2020, doi:10.3390/gels6030029_

Round 1
Reviewer 1 Report
This review written by one of the pioneers in the field of cryogels covers more than
40-years of research in the field in the A.N.Nesmeyanov Institute of Organoelement Compounds. It presents not only main key points and achievements made up to know but results of ongoing projects. The review is very well organized, thoughtfully written with minimum typos. In my opinion this review can be accepted for publication after minor revision.
- Classification of cryostructurates ( p.1 Lines 37-49) and cryotropic gels (p.14, lines 16-30) are placed in different sections and separated with two long tables, no reference is given in the Introduction for the further text.
- I would prefer to see subsections in sections 3.1. and 3.2 instead of “First:,Second:, and so on). This will give additional focus to a reader and allow to find required information without going through the whole review
- Subheadings instead of i, ii et cet would be helpful in section 3.3. Subsections can be also added to facilitate reading
- Figure 4:. Probably a fragment in the heading “……with respect of the devel.” In my opinion, scaffolds with human cells shall be linked with Biomedical Material rather than with Biotechnological Materials, as it is shown now.
- 19 line 180: probably growth not gross
Author Response
Reviewer 1
This review written by one of the pioneers in the field of cryogels covers more than 40-years of research in the field in the A.N.Nesmeyanov Institute of Organoelement Compounds. It presents not only main key points and achievements made up to know but results of ongoing projects. The review is very well organized, thoughtfully written with minimum typos. In my opinion this review can be accepted for publication after minor revision.
Thanks a lot for your positive, in whole, evaluation of this article.
- Classification of cryostructurates ( p.1 Lines 37-49) and cryotropic gels (p.14, lines 16-30) are placed in different sections and separated with two long tables, no reference is given in the Introduction for the further text.
The following notes are added in this paragraph of the revised manuscript.
In turn, the nodes of the three-dimensional polymer network of such cryogels can be stabilized by covalent, ionic, coordination and non-covalent bonds or a combination of them. A more detailed discussion of these systems is given in section 2.
- I would prefer to see subsections in sections 3.1. and 3.2 instead of “First:,Second:, and so on). This will give additional focus to a reader and allow to find required information without going through the whole review
The reviewers’ notes: “The review is very well organized …” (Reviewer 1); “The paper is well written and arranged” (Reviewer 2); “This paper is very well documented and well organized …” (Reviewer 3). On the other hand, the recommendation of the reviewer 3 is to insert the additional subsections (?). Nonetheless, I have tried to change “First:,Second:, and so on” for the some subsections like the 3.1.1; 3.2.3 and so on. The result turned out to be unsatisfactory – the text became excessively "motley"; moreover, it was impossible to create too short subsections. Therefore, I prefer to retain the original version of the text in sections 3.1 and 3.2.
- Subheadings instead of i, ii et cet would be helpful in section 3.3. Subsections can be also added to facilitate reading
The same comments as in the above case.
- Figure 4:. Probably a fragment in the heading “……with respect of the devel.”
Necessary words disappeared obviously upon the re-formatting in the journal. The heading should be as follows: “Schematic diagram showing the applied R&D activities of the researches with respect of the developed in IOEC cryogenically-structured materials” (it was corrected in the revised version).
In my opinion, scaffolds with human cells shall be linked with Biomedical Material rather than with Biotechnological Materials, as it is shown now.
Of course, the bioartificial constructs “scaffolds with human cells” could be attributed to the Biomedical Materials, as well, but providing the constructs are inserted in the respective organisms. However, the studies performed with the participation of the researchers from IOEC mainly dealt with the preparation of the respective scaffolds and their biotesting in vitro, that is, really upon the biotechnological experiments rather than the biomedical ones. Therefore, these works were included in the ‘Biotechnological Materials’ of the diagram Fig.4.
- 19 line 180: probably growth not gross
Unfortunately, the ‘spell check’ option did not find such fragment in the manuscript, where the word ‘gross’ is used only for the size of pores.
Reviewer 2 Report
The manuscript by Lozinsky reviews the contribution of a research institute to the field of polymeric cryogels. The article describes the results of more than 40-years of studies in this field. A brief historical information, a description of the main peculiarities of the cryostructuring processes, morphological features of polymeric cryogels, and examples of their application are involved. The paper is well written and arranged. It can be accepted for publication without corrections except the Conclusion section. The last does not exactly correspond to the core content of the paper. Also, including references in this part is not recommended.
Author Response
Reviewer 2
The manuscript by Lozinsky reviews the contribution of a research institute to the field of polymeric cryogels. The article describes the results of more than 40-years of studies in this field. A brief historical information, a description of the main peculiarities of the cryostructuring processes, morphological features of polymeric cryogels, and examples of their application are involved. The paper is well written and arranged. It can be accepted for publication without corrections except the Conclusion section. The last does not exactly correspond to the core content of the paper. Also, including references in this part is not recommended.
Thank you for your positive review of this article.
As for the comments on the "Conclusion" section:
Although I have published more than a dozen reviews in General, I have never encountered any canonical constructions that are included in the rules for authors about the content of such a section specifically for review articles. This requirement is also absent in the rules for authors of the ‘Gels’ journal. An experimental article is another matter. In the latter case, the "Conclusion" section summarizes what has been done and what the main results are. In the case of the present review, the issues raised in it are very diverse, and they are discussed in the text of the article itself.
Regarding the presence of references in the "Conclusion" section:
It is clear that mentioning new reviews published when the work on this article was actually finished without quotation of these just-published articles would look rather strange. On the other hand, I tried to remove the references that illustrate the interest in the direction of "Cryostructuring" on different continents, but then the statement of this interest turns out to be quite unsubstantiated. In this regard, I think the presence of the respective references in the "Conclusion" section of this review article is quite justified.
Reviewer 3 Report
The paper is very well documented and well organized underlining the enormous efforts of the researchers to develop this scientific area. The manuscript in easy to read being useful for beginners in the field; the author is presenting the basic information about cryogels along with the physical–chemical mechanism of cryogels formation and their characteristics regarding this aspect. I consider the manuscript is well written and ready to be published and I congratulate the author on his work!
Author Response
Reviewer 3
The paper is very well documented and well organized underlining the enormous efforts of the researchers to develop this scientific area. The manuscript in easy to read being useful for beginners in the field; the author is presenting the basic information about cryogels along with the physical–chemical mechanism of cryogels formation and their characteristics regarding this aspect. I consider the manuscript is well written and ready to be published and I congratulate the author on his work
Thank you very much for your appreciation of this review, which took from the author a lot of effort.